# Teaching Language Models to Reason with Tools

**Chengpeng Li**[*1,2], **Zhengyang Tang**[*2,3], **Ziniu Li**[*3,4], **Mingfeng Xue**[2], **Keqin Bao**[1,2], **Tian Ding**[4],
**Ruoyu Sun**[3,4], **Benyou Wang**[3], **Xiang Wang**[1], **Junyang Lin**[2], and **Dayiheng Liu**[†2]

[1]University of Science and Technology of China
[2]Qwen Team, Alibaba Inc.
[3]The Chinese University of Hong Kong, Shenzhen
[4]Shenzhen International Center for Industrial and Applied Mathematics, Shenzhen Research Institute
of Big Data

## Abstract

Large reasoning models (LRMs) like OpenAI-o1 have shown impressive capabilities in natural language reasoning. However, these models frequently demonstrate inefficiencies or inaccuracies when tackling complex mathematical operations. While integrating computational tools such as Code Interpreters (CIs) offers a promising solution, it introduces a critical challenge: a conflict between the model's internal, probabilistic reasoning and the external, deterministic knowledge provided by the CI, which often leads models to unproductive deliberation. To overcome this, we introduce CoRT (Code-Optimized Reasoning Training), a post-training framework designed to teach LRMs to effectively utilize CIs. We propose *Hint-Engineering*, a new data synthesis strategy that strategically injects diverse hints at optimal points within reasoning paths. This approach generates high-quality, code-integrated reasoning data specifically tailored to optimize LRM-CI interaction. Using this method, we have synthesized 30 high-quality samples to post-train models ranging from 1.5B to 32B parameters through supervised fine-tuning. CoRT further refines the multi-round interleaving of external CI usage and internal thinking by employing rejection sampling and reinforcement learning. Our experimental evaluations demonstrate CoRT's effectiveness, yielding absolute improvements of 4% and 8% on DeepSeek-R1-Distill-Qwen-32B and DeepSeek-R1-Distill-Qwen-1.5B, respectively, across five challenging mathematical reasoning datasets. Moreover, CoRT significantly enhances efficiency, reducing token usage by approximately 30% for the 32B model and 50% for the 1.5B model compared to pure natural language reasoning baselines. The models and code are available at: https://github.com/ChengpengLi1003/CoRT.

## 1 Introduction

Benefiting from advancements in reinforcement learning (RL) techniques [1–4], Large Reasoning Models (LRMs) such as OpenAI-o1 [5], Qwen-3 [6], and DeepSeek-R1 [7] have achieved breakthrough progress in complex reasoning tasks [8, 9]. These models exhibit numerous human-like cognitive strategies with long chain of thought (CoT) [10, 11] reasoning, including self-refinement, self-reflection, and multi-strategy exploration. However, LRMs still demonstrate limitations in accuracy and efficiency when handling complex mathematical operations, such as precise computation

---

∗: Equal contribution. Email: chengpengli@mail.ustc.edu.cn, zhengyangtang@link.cuhk.edu.cn, ziniuli@link.cuhk.edu.cn

†: Corresponding author.

39th Conference on Neural Information Processing Systems (NeurIPS 2025).

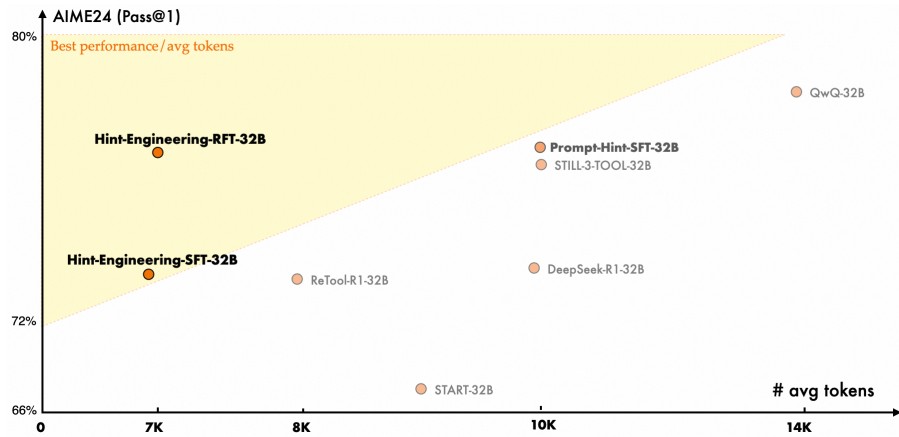

Figure 1: Performance vs. token efficiency on AIME24. The x-axis represents average token usage while the y-axis shows Pass@1 accuracy. Hint-Engineering-RFT-32B achieves comparable accuracy to other frontier models while using significantly fewer tokens.

and complex equation solving [12, 13], which are better suited for code interpreters (CIs). Leveraging CIs, LRMs like o3 and o4-mini [14] have substantially enhanced their mathematical reasoning capabilities. However, as these models are proprietary, the methods for achieving this effective LRM-CI integration remain undisclosed.

A key open challenge is teaching LRMs when and how to effectively and efficiently use CIs to generate structured reasoning. This is a scientifically new problem because, unlike pure natural language reasoning, CIs introduce *external deterministic* knowledge that exists beyond the model's internal representations. This raises critical questions: (1) How can we synthesize high-quality training data when models like o3 and o4-mini do not expose their detailed reasoning traces? (2) How can the model effectively coordinate a CI's computational precision with its abstract CoT reasoning capabilities? (3) How can the self-reflection mechanisms inherent to LRMs be reconciled with the exact external knowledge provided by CIs?

This paper explores to answer the above questions. We begin by tackling the data synthesis challenge, which forms the foundation for post-training via supervised fine-tuning (SFT) [15], rejection fine-tuning (RFT) [16] and RL [7]. Based on the open-source LRM DeepSeek-R1, we investigate direct prompting methods like [17] for CI integration. Our key discovery is that inserting a simple hint—"Okay, let's try to solve this problem step by step using multiple python code calls"—immediately after the model's thinking token <think> improves code triggering rates from 50% to 90% (on 100 problems from [17]). We term this straightforward approach the *prompt-hint* method. This confirms that LRMs possess the latent capability to leverage CIs for reasoning despite being primarily trained on natural language. However, we also find that they struggle with efficient tool utilization. The two most prominent inefficiencies are **delayed code computation** (preferring text reasoning before utilizing a CI) and **code result distrust** (unnecessarily verifying CI outputs manually). To address these limitations, we have designed a more sophisticated approach we call *hint-engineering*. The key idea is to strategically insert different hints at appropriate positions throughout the reasoning process. These hints are specifically designed to teach the LRM to understand the outputs of CIs, mitigating meaningless reflection behaviors. A notable feature of this approach is that the resulting reasoning traces become significantly shorter and more efficient.

Following the principle that data quality outweighs quantity (less is more) [18–20], we manually generate 30 high-quality samples with human verification. Using these samples, we post-train models of varying sizes based on available computational resources. For large 32B parameter models, we conduct SFT and RFT, while RL remains computationally infeasible within our infrastructure. However, we successfully implement the complete SFT-RFT-RL pipeline for smaller models. Moreover, we carefully design outcome rewards to encourage correct code generation behaviors.

Our experiments confirm the effectiveness of the above approaches. Results across five challenging mathematical reasoning datasets demonstrate that Hint-Engineering models achieve significant improvements: 4% absolute accuracy gain for DeepSeek-R1-Distill-Qwen-32B and 8% for DeepSeek-

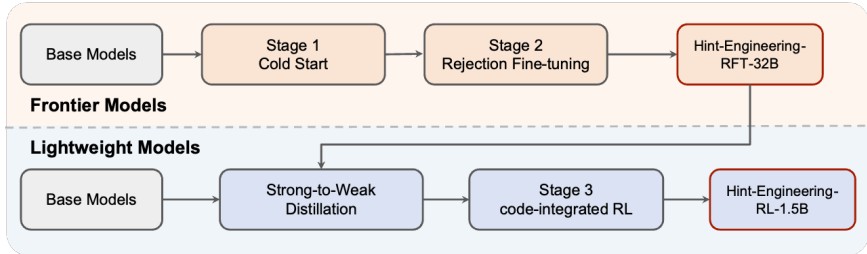

Figure 2: The training framework of CoRT.

R1-Distill-Qwen-1.5B. Moreover, on the most challenging AIME benchmarks, our approach reduces token consumption by 30% for the 32B model and 50% for the 1.5B model.

To summarize, our key contributions include:

- A new data synthesis framework specifically engineered for code-integrated reasoning that effectively addresses the critical data scarcity challenge in this emerging domain.
- An efficient and scalable training pipeline that enables LLMs to acquire sophisticated code-integrated reasoning capabilities through targeted post-training procedures.
- Comprehensive empirical evaluations demonstrating significant performance and token efficiency improvements across 5 challenging mathematical benchmarks. We further validate the generalizability of our approach with an out-of-distribution evaluation on chemistry problems (Appendix I).

## 2 Methodology

This section introduces the CoRT framework (Figure 2), which involves three key stages: modeling code-integrated reasoning, training 32B models (Cold Start, SFT, RFT), and developing 1.5B models through strong-to-weak distillation and RL.

### 2.1 Task Formulation

By leveraging executable programs, LRMs can now perform precise calculations and complex logical operations. The framework comprises three essential components: a problem input $P$, a language model $\pi$, and an executor environment $\mathcal{E}$. During the reasoning process, the system constructs a sequence $\tau_t$ at time step $t$, which can be represented as:

$$\tau_t = \{(n_1, p_1, o_1), \ldots, (n_t, p_t, o_t)\}. \tag{1}$$

Here, $n_i$ represents the textual reasoning step, $p_i$ denotes the program snippet generated by the model, $o_i$ indicates the execution output, with $i$ indexing the sequential interactions between the language model and the execution environment. The sequential reasoning process follows these steps:

$$
\begin{aligned}
(t_t, p_t) &= \pi(P \oplus \tau_{t-1}), o_t = \mathcal{E}(p_t), \\
\tau_t &= \tau_{t-1} \oplus n_t \oplus p_t \oplus o_t.
\end{aligned}
\tag{2}
$$

This iterative mechanism establishes a dynamic feedback loop, where each reasoning step is informed by previous computational results. The process continues until the model reaches a definitive answer.

### 2.2 Cold Start Methods

#### 2.2.1 Prompt-hint

This section explores a straightforward approach, referred to as prompt-hint, for constructing a cold-start dataset for teaching LRMs. Specifically, to initiate our data generation process, we carefully crafted a prompt (detailed in Appendix E) designed to instruct R1 [7] to leverage both natural language reasoning and interactive Python code execution during inference. We integrated a code interpreter that enables R1 to perform real-time interactive reasoning, as outlined in Section 2.1. To encourage generating reasoning trajectories that incorporate code, inspired by [21], we enhanced the generation process by introducing a strategic hint—"Okay, let's try to solve this problem step by step using multiple python code calls"—following the model's initial thinking token <think>. An example is shown in Figure 3 (a).

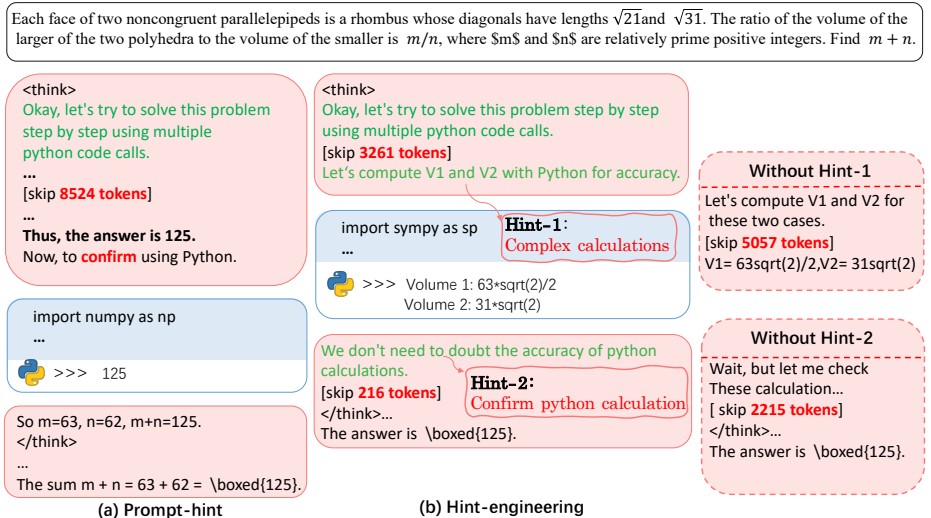

Figure 3: Comparison between prompt-hint and hint-engineering approaches using Problem 13 from AIME23 I as a case study (prompt prefix omitted for brevity).Both methods begin with a general hint (in green) after <think> to encourage code usage. While prompt-hint (a) allows natural interaction between R1 and the Code Interpreter (CI), leading to inefficient token usage, hint-engineering (b) introduces strategic hints at key decision points. Hint-1 is inserted when the model begins manual calculation of complex volumes (V1 and V2), redirecting to Python computation. Hint-2 is added to prevent unnecessary verification of Python calculations. Through these targeted interventions, hint-engineering achieves approximately 5000 tokens reduction while maintaining solution accuracy. More examples are provided in Appendix F.

Leveraging the publicly available STILL3 [17] dataset comprising 820 math problems and R1, we employed our prompt-hint annotation method to generate 800 training instances, denoted as $D_{\text{prompt-hint}}$. We then performed SFT with diversity preserving [15] on DeepSeek-R1-Distill-Qwen-32B using this dataset, resulting in our Prompt-Hint-SFT-32B model.

### 2.2.2 Hint-engineering

Despite the simplicity of the prompt-hint approach, where LRMs autonomously decide when and how to utilize the Code Interpreter (CI), we find it does not fundamentally address the challenge of interleaving internal probabilistic reasoning with external deterministic knowledge. Specifically, we identified several inefficiencies and instances of overthinking. These limitations can be categorized into two main issues:

- **Delayed code computation**: When handling complex mathematical operations, models tend to first engage in text-based reasoning before writing code and using CI for verification. This pattern often results in redundant computational steps.

- **Code result distrust**: Upon receiving CI execution results, models frequently display a lack of trust in the output, leading to unnecessary manual verification and redundant calculations.

These behavioral patterns significantly impact the model's reasoning efficiency, particularly in terms of the number of tokens required for problem-solving.

To address these inefficiencies, we implement a targeted approach named hint-engineering. When delayed computation is detected—specifically, at the point where the model begins to manually calculate complex mathematical operations—we insert a strategic hint, such as, "It looks tedious, and we can use python code to simplify the reasoning", followed immediately by a Python code trigger ```python. Similarly, when code result distrust emerges, we introduce a hint such as "We don't need to doubt the accuracy of python calculations". This intervention redirects the model's focus back to the core problem rather than engaging in unnecessary verification of computational accuracy. It is important to note that while we discourage the verification of Python's numerical calculations, we maintain the model's behavior to verify the logical correctness of the code structure. Figure 3 (b) illustrates a concrete example. We formulate this process in Appendix B.

A critical challenge in our approach was identifying suitable positions for hint insertion. While we initially attempted to automate this process using DeepSeek-V3 and R1, we found the results to be insufficiently precise. Hence, we opted for manual hint insertion on 30 problems from AIME (pre-2024) to create the $D_{\texttt{Hint-engineering-SFT}}$ dataset and train the Hint-Engineering-SFT-32B model.

To further enhance model performance, we conducted rejection fine-tuning(RFT) using the Hint-Engineering-SFT-32B model on the 820 problems from STILL3. Specifically, we performed multiple sampling iterations on each problem and implemented a filtering process to eliminate trajectories with incorrect final answers, as well as those exhibiting delayed code computation or code result distrust behaviors. We combined the filtered trajectories with $D_{\texttt{Hint-engineering-SFT}}$ to create $D_{\texttt{Hint-engineering-RFT}}$, a dataset of 830 examples. This curated dataset was then used to fine-tune DeepSeek-R1-Distill-Qwen-32B, resulting in our Hint-Engineering-RFT-32B model.

### 2.3 Strong-to-weak Distillation

We aim to use RL to refine the multi-round interleaving of CI usage and internal thinking, as RL allows models to interact directly with the CI. However, computational constraints made it infeasible to perform RL on 32B-parameter models. We therefore opted for a strong-to-weak distillation approach. We distilled both the Prompt-Hint-SFT-32B and Hint-Engineering-RFT-32B models into the DeepSeek-R1-Distill-Qwen-1.5B architecture. This process yielded two smaller models, Prompt-Hint-1.5B-SFT and Hint-Engineering-SFT-1.5B, which served as the foundation for our RL experiments. We selected and preprocessed 10k examples from publicly available datasets for this distillation, with detailed procedures provided in Appendix G.

### 2.4 Code-integrated Reinforcement Learning

To further refine the models' code-integrated reasoning, we apply RL to the 1.5B models. Following recent trends in value-free methods [2], we chose the GRPO algorithm [3], as it is well-suited for scenarios with sufficient rollouts. In applying the GRPO algorithm to our models, we introduced several modifications to the standard text-based GRPO framework:

- **Rollout with Code Interpreter**: We enable multiple model-CI interactions during the RL rollout process, as described in Section 2.1. To manage computational overhead during rollouts, we implement a maximum tool usage limit $T$. Once this limit is reached, we append a hint informing the model to proceed without further Python usage.

- **Persistent Execution Environment**: Unlike traditional TIR environments that execute each Python block independently, we construct a Jupyter-like environment where variables, environments, and functions persist across code blocks, enhancing code efficiency and reducing errors.

- **Output Masking**: To ensure training stability, we implement execution result masking, significantly reducing model collapse probability during training. This crucial modification prevents potential training failures that would otherwise occur without such masking.

- **Reward Design**: We implement a dual reward system comprising accuracy reward and code execution reward as defined in Equation (3). For accuracy assessment, we require models to present final answers in a specified format (e.g., within boxed{}), enabling reliable rule-based verification against ground truth answers. To prevent infinite loops resulting from repeated code failures, we implement a code execution penalty for responses where all code execution attempts fail. The total reward R is computed as a weighted sum of these two components, where $\omega$ controls the contribution of the code execution penalty.

$$R_a = \begin{cases} 1 & \text{if answers match} \\ 0 & \text{otherwise} \end{cases} \qquad R_c = \begin{cases} -1 & \text{if all codes fail} \\ 0 & \text{otherwise} \end{cases} \qquad R = R_a + \omega R_c \quad (3)$$

## 3 Experiments

In this section, we evaluate the effectiveness of our proposed CoRT framework through comprehensive experiments on five challenging mathematical reasoning benchmarks: (1) AIME24, (2) AIME25,(3) AMC23, (4) MATH500, and (5) OlympiadBench. Comprehensive descriptions of the evaluation datasets are provided in Appendix C. For space saving, our implementation details of SFT, RFT, RL and inference are listed in Appendix A. Due to space constraints, we present only representative experimental results in the main text, with comprehensive results available in Appendix D. Moreover, the baseline models are described in Appendix H.

## 3.1 Main Results

Table 1: Performance comparison of different math reasoning models across benchmarks. For each section, best results are shown in **bold** and second-best results are underlined. During inference, we set temperature 0.6 and top$_p$ 0.95. Results for AIME24, AIME25, and AMC23 are averaged over 16 samples, while MATH500 and Olympiad results are averaged over 4 samples. All experiments use a maximum sequence length of 32,768 tokens and limit tool usage to 15 calls.

| Model | Tool-Use | Stage | AIME24 | AIME25 | AMC23 | MATH500 | Olympiad | Avg |
|---|---|---|---|---|---|---|---|---|
| *SOTA Models* | | | | | | | | |
| o1 | ✗ | RL | 74.3 | **79.2** | - | 96.4 | - | - |
| o3 | ✓ | RL | 95.2 | **98.7** | - | - | - | - |
| o4-mini | ✓ | RL | 98.4 | 99.5 | - | - | - | - |
| DeepSeek-R1 | ✗ | RL | **79.8** | 70.0 | - | 97.3 | - | - |
| QwQ-32B | ✗ | RL | 79.5 | 65.3 | **94.3** | 92.3 | **79.7** | **82.2** |
| *Frontier Models (32B)* | | | | | | | | |
| DeepSeek-R1-32B | ✗ | SFT | 72.9 | 59.0 | 88.8 | 94.3 | 72.5 | 77.5 |
| START-32B | ✓ | SFT | 66.7 | 47.1 | **95.0** | 94.4 | - | - |
| STILL-3-TOOL-32B | ✓ | SFT | 76.7 | 64.4 | 91.3 | **96.6** | **75.9** | 81.0 |
| ReTool-R1-32B | ✓ | RL | 72.5 | 54.3 | 92.9 | 94.3 | 69.2 | 76.6 |
| Prompt-Hint-SFT-32B | ✓ | SFT | **77.3** | 65.0 | **95.0** | **96.6** | 75.1 | **81.8** |
| Hint-Engineering-SFT-32B | ✓ | SFT | 72.1 | 60.2 | 91.3 | 94.4 | 71.2 | 77.8 |
| Hint-Engineering-RFT-32B | ✓ | RFT | 76.7 | **67.1** | 94.4 | 95.1 | 73.4 | 81.3 |
| *Lightweight Models (1.5B)* | | | | | | | | |
| DeepSeek-R1-1.5B | ✗ | SFT | 28.8 | 21.8 | 62.9 | 83.9 | 43.3 | 48.1 |
| DeepScaleR-1.5B-Preview | ✗ | RL | 40.0 | 30.0 | 73.6 | **87.8** | 50.0 | 56.3 |
| ToRL-1.5B | ✓ | RL | 26.7 | 26.7 | 67.5 | 77.8 | 44.0 | 48.5 |
| Prompt-Hint-1.5B-SFT | ✓ | SFT | 30.6 | 25.0 | 63.1 | 83.3 | 50.4 | 50.5 |
| Prompt-Hint-1.5B-RL | ✓ | RL | **43.1** | **30.2** | **73.8** | 87.3 | **57.1** | **58.3** |
| Hint-Engineering-1.5B-SFT | ✓ | SFT | 34.0 | 23.5 | 64.6 | 84.2 | 49.8 | 51.2 |
| Hint-Engineering-1.5B-RL | ✓ | RL | 41.0 | 29.4 | 70.0 | 85.8 | 55.6 | 56.4 |

Table 1 presents our main results, comparing our models with state-of-the-art baselines across multiple mathematical reasoning benchmarks. We organize the results into three sections: SOTA Models, Frontier Models (32B), and Lightweight Models (1.5B).

For 32B models, we observe that after SFT, our models achieve performance comparable to existing tool-integrated models, with Prompt-Hint-SFT-32B slightly outperforming others with an average accuracy of 81.8% across benchmarks. Notably, Hint-Engineering-RFT-32B, despite being trained on just 30 manually annotated examples initially, achieves competitive performance with an average accuracy of 81.3%. This highlights the effectiveness of our rejection fine-tuning approach and the importance of high-quality data over quantity.

For 1.5B models, the reinforcement learning stage brings substantial improvements. Prompt-Hint-1.5B-RL achieves state-of-the-art performance among lightweight models with an average accuracy of 58.3%, outperforming the non-tool-using DeepScaleR-1.5B-Preview. Similarly, Hint-Engineering-1.5B-RL shows strong performance at 56.4%. The dramatic improvement from SFT to RL stages (approximately 8% absolute gain) demonstrates the effectiveness of our reinforcement learning approach for tool-integrated reasoning.

## 3.2 Token Efficiency Analysis

Beyond raw performance, we analyze the token efficiency of our models. Token efficiency can be roughly estimated by dividing the model's accuracy by its average token consumption. Figures 1 and 4 illustrate this analysis, revealing several key insights:

- **Superior Efficiency of Hint-Engineering**: At equivalent performance levels, Hint-Engineering series models demonstrate the highest token efficiency. For example, as shown in Figure 1, Hint-Engineering-RFT-32B achieves the same performance as QwQ-32B while using 50% fewer tokens (7K vs 14K). Comparing Hint-Engineering-SFT-32B with R1-distill-32B, with just 30 training examples for fine-tuning, the model reduces token consumption by approximately 30% while maintaining comparable performance. As observed from the inference token budget analysis in Figure 4 (a), Hint-Engineering achieves superior performance compared to Prompt-Hint under

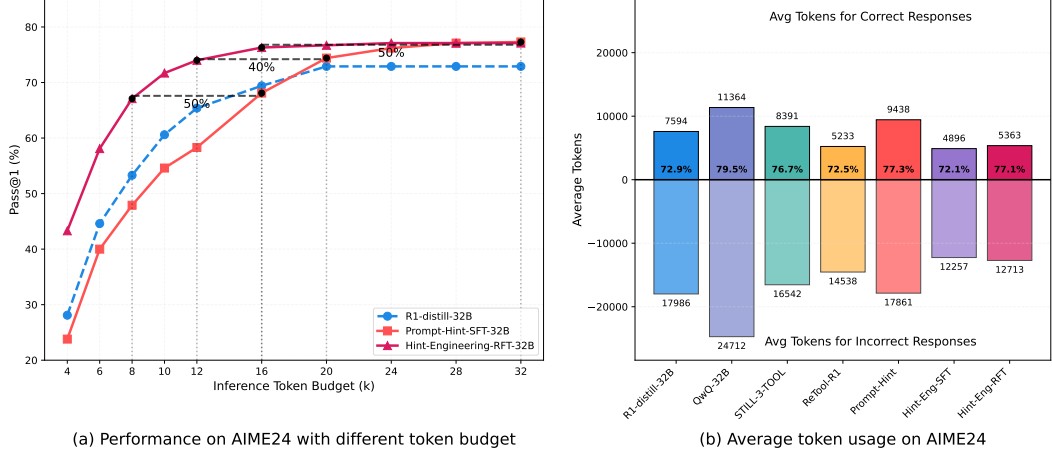

(a) Performance on AIME24 with different token budget  (b) Average token usage on AIME24

Figure 4: Token efficiency analysis on AIME24. (a): Token efficiency comparison showing Hint-Engineering-RFT-32B achieves comparable accuracy with significantly fewer tokens (40-50% token saving) compared to Prompt-Hint-SFT-32B. (b): Average token usage for correct and incorrect responses across different models, with Hint-Engineering models maintaining lower token consumption while achieving competitive performance.

limited token budgets, while Prompt-Hint shows no significant advantage over CoT in these constrained conditions.

- **Low Token Usage for both Correct and Incorrect Responses**: As shown in Figure 4 (b), compared to CoT and Prompt-Hint approaches, Hint-Engineering reduces token consumption in both correct and incorrect responses, indicating that it not only solves problems efficiently but also minimizes token waste during unsuccessful attempts. This improvement stems from Hint-Engineering's fundamental design: increasing the utilization of code for computations and enhancing confidence in code execution results.

As shown in Figure 1, when plotting performance against token usage, Hint-Engineering-RFT-32B sits in the optimal region with high performance and low token consumption, demonstrating the best performance-to-efficiency ratio among all compared models.

## 3.3 Code Behavior Analysis between Prompt-Hint and Hint-Engineering

We first establish a taxonomy for Python code usage based on two dimensions. From the perspective of reasoning relationship, we categorize code usage into **Calculation** (computing results not present in the current reasoning chain) and **Verification** (validating results derived from chain-of-thought reasoning). In terms of specific functionality, we classify code into categories including Solving Equations, Numerical Approximation, Pattern Recognition, Combinatorial Enumeration and so on.

We conduct a comprehensive analysis of Python code usage patterns across all test sets, comparing Prompt-Hint-SFT-32B and Hint-Engineering-RFT-32B, two models with comparable overall performance. We employ DeepSeek-V3 to classify Python code functionality, with the corresponding classification prompts detailed in Appendix E.

Figure 5 provides a qualitative analysis of the code behavior patterns in our different approaches. The most striking difference is in how the models utilize code:

- **Prompt-Hint Approach**: Code is predominantly used for verification purposes (68.2%), with only 31.8% dedicated to actual computational tasks. This indicates an inefficient utilization pattern where the model performs calculations in natural language and then uses code primarily to verify these calculations.

- **Hint-Engineering Approach**: Shows a much more balanced usage, with 51.1% of code dedicated to direct calculation and 48.9% for verification. This more optimal distribution reflects the model's understanding of when to leverage computational tools versus when to rely on reasoning.

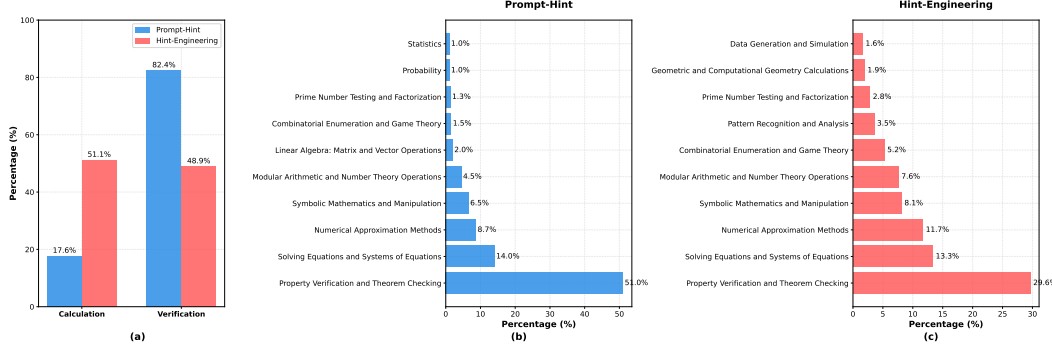

Figure 5: Analysis of Python code usage patterns. (a) Distribution of code usage types: Hint-Engineering shows a preference for calculation while Prompt-Hint favors verification tasks. (b) and (c) Function-specific distribution: Hint-Engineering demonstrates more balanced usage across different Python functions compared to Prompt-Hint.

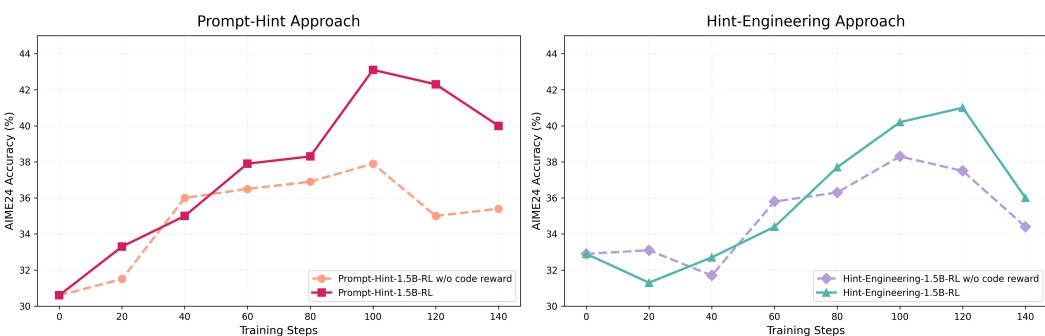

Figure 6: Ablation study on the impact of code execution reward during RL training on AIME24. Left: Performance of Prompt-Hint-1.5B-RL with and without code reward. Right: Performance of Hint-Engineering-1.5B-RL with and without code reward. Both approaches show consistent performance improvements when trained with the additional code execution reward.

Additionally, the Hint-Engineering approach shows greater diversity in the types of computational operations performed, including symbolic mathematics, equation solving, and combinatorial enumeration. This suggests that the model has developed a more sophisticated understanding of the appropriate use cases for different types of code operations.

As shown in Figure 5, Prompt-Hint and Hint-Engineering exhibit distinct code-integrated Reasoning patterns. Prompt-Hint demonstrates a strong preference for verification (82.4%), while Hint-Engineering maintains a relatively balanced distribution between calculation and verification (approximately 50% each). This balanced distribution emerges from the interplay between our Hint-Engineering design, which encourages computational efficiency, and the model's inherent tendency toward verification in long chain-of-thought reasoning. Regarding specific mathematical functionalities, we observe that Property Verification and Theorem Checking dominates Prompt-Hint's code usage (51%), whereas Hint-Engineering exhibits a more uniform distribution across different functions. Interestingly, both approaches share the same top-5 most frequently used Python functions, suggesting the influence of the test sets' mathematical domains. Moreover, we present representative examples in Appendix F.

### 3.4 Impact of Code Reward in RL

Figure 6 presents an ablation study on the effect of incorporating code execution reward into the RL process. We set the code reward ratio $\omega = 0.1$ here. The results demonstrate that incorporating this code reward consistently improves performance for both approaches:

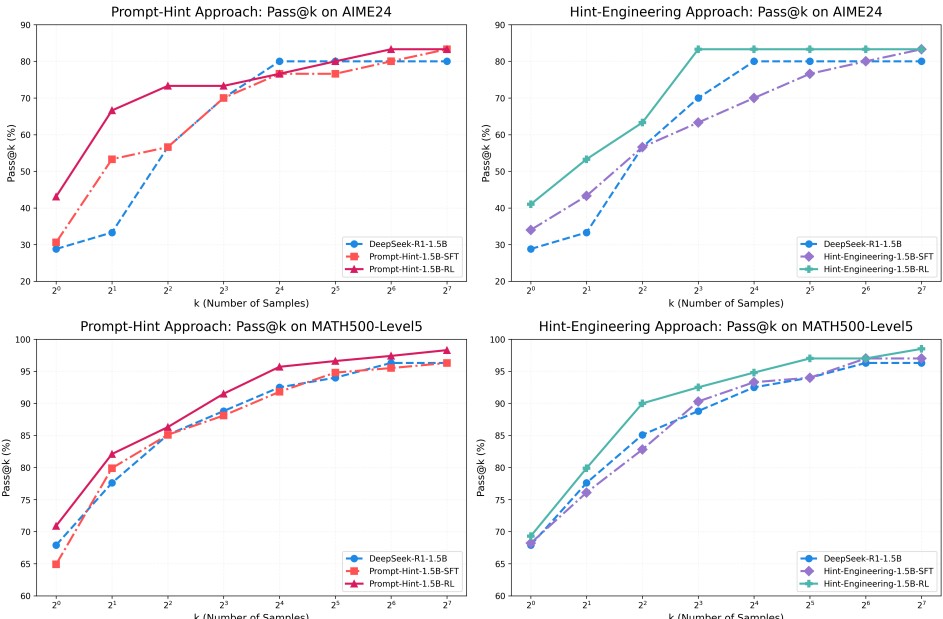

Figure 7: Pass@k performance on AIME24 (top) and MATH500-Level5 (bottom) for both Prompt-Hint (left) and Hint-Engineering (right). The analysis compares the base model DeepSeek-R1-1.5B with SFT and RL variants. While SFT does not significantly improve the Pass@k upper bound, RL substantially elevates performance across all k values, particularly at lower sampling budgets.

- **For Prompt-Hint**: Models trained with code reward achieve up to 5% higher accuracy than those without, reaching a peak of 43.1% versus 37.9%.

- **For Hint-Engineering**: A similar pattern emerges with approximately 3% performance improvement, reaching 41.0% versus 38.3%.

Notably, we found that the magnitude of this reward is crucial: a modest code reward ratio $\omega = 0.1$ provides optimal results, while stronger penalties (e.g., 0.5) degraded performance. This suggests that while encouraging code correctness is valuable, overly penalizing experimental code attempts can inhibit the model's exploration and learning. Additional experimental results can be found in Appendix D.

## 3.5 Pass@K Analysis

Figure 7 illustrates the performance of our models as a function of sample size ($k$) on both AIME24 and MATH500-Level5 datasets. Several important patterns emerge:

- **SFT Impact on Reasoning Ceiling**: Supervised fine-tuning alone does not significantly raise the Pass@k upper bound for either approach. This suggests that while SFT can teach the model format and basic tool usage, it doesn't fundamentally enhance the model's reasoning capabilities for 1.5B size model with the selected 10k probelms.

- **RL Significantly Raises Performance Ceiling**: Both Prompt-Hint-1.5B-RL and Hint-Engineering-1.5B-RL show substantially higher Pass@k curves than their SFT counterparts, particularly at lower k values. This indicates that reinforcement learning successfully improves not just the average performance but the model's ability to consistently arrive at correct solutions with fewer attempts.

These observations confirm that RL effectively amplifying the benefits of the more optimal code usage patterns established during the Hint-Engineering training.

Additional interesting findings, such as the impact of problem difficulty on RL performance and the evolution of code behavior during RL training, are documented in Appendix D.

## 4  Related Work

### 4.1  Reasoning in LLMs

The evolution of reasoning capabilities in LLMs has progressed rapidly in recent years [22–24, 3, 5, 7, 6, 14]. For comprehensive coverage of this field, we direct readers to recent surveys [8, 9, 25–27]. A pivotal technique in this field is Chain-of-Thought (CoT) reasoning [10], which, when combined with Transformer architectures [28], enables models to perform complex computational tasks [29]. This field has benefited substantially from scaling up synthetic data [30–32]. These advances leverage various approaches, including preference optimization [33], tree search [34, 35], and advanced exploration strategies [36, 37].

Notably, following OpenAI's o1 [38], there has been a significant trend towards long-form CoT reasoning, incorporating human-like cognitive patterns such as multiple reflections, task decomposition, and strategic exploration and these LLMs are often called large reasoning models (LRMs). This integration of human-inspired reasoning approaches has led to substantial improvements in complex reasoning tasks, particularly in coding and mathematics, where models have demonstrated unprecedented capabilities in systematic problem-solving [39–42, 36]. While natural language remains the primary medium for reasoning in this domain, there is growing interest in integrating external tools such as code interpreters [43, 44, 12] and automatic verification systems [45–47] to overcome the inherent limitations of natural language reasoning, such as accurate computation.

### 4.2  Code-Integrated Reasoning for LLMs

Recent research has explored integrating external tools with language models to enhance their reasoning capabilities [48–53]. ToRA [12] pioneered code-integrated reasoning specifically for mathematical problem-solving, demonstrating that offloading complex calculations to specialized systems significantly improves performance. Since then, COA [54] trains LLMs to decode reasoning chains with abstract placeholders, and then call domain tools to reify each reasoning chain by filling in specific knowledge. rStar-Math [55] introduced a code-augmented Chain of Thought data synthesis method through Monte Carlo Tree Search. Recent works [56–59] have initiated investigations into frameworks that enable base models to autonomously develop code-integrated reasoning capabilities for mathematical problem-solving.

With the advancement of LRMs, the organic integration of code interpreters within long-form CoT reasoning has emerged as a crucial research challenge. Several concurrent works have explored this direction. START [21], which shares similarities with our approach, introduces hints to guide code generation within large reasoning modes; however, their random hint insertion may lead to suboptimal utilization of code interpreters. STILL3 [17] employs prompting techniques to construct code-integrated data, similar to our proposed Prompt-Hint method, but does not address reasoning efficiency. Retool [60] attempts to bootstrap training data by rewriting long CoT reasoning, yet shows limited performance improvements when based on DeepSeek-R1-Distill-Qwen-32B. While OTC [59] considers efficiency from the perspective of tool call frequency, it does not explore methods for enhancement building upon existing LRMs. CoRT proposed a highly sample-efficient approach that achieved both performance breakthroughs and significant improvements in reasoning efficiency. Through human-in-the-loop annotation of 30 high-quality samples, combined with techniques such as RFT [16, 61] and RL, they demonstrated that substantial improvements in both reasoning capabilities and efficiency can be achieved with minimal high-quality training data.

## 5  Conclusion

Our experiments reveal that properly integrated code tools enhance mathematical reasoning across model scales. High-quality data with optimal code behavior patterns can match or exceed the performance of larger datasets, while reinforcement learning significantly improves performance beyond SFT, particularly for smaller models. The Hint-Engineering approach achieves remarkable efficiency, reducing token usage by 30-50% while maintaining competitive performance. Moreover, RL shapes code usage behavior toward either efficiency or increased integration. These findings demonstrate that combining high-quality data curation, targeted fine-tuning, and reinforcement learning with carefully designed rewards effectively enhances mathematical reasoning capabilities through tool integration.

## Acknowledgments

This work of Xiang Wang is supported by the National Natural Science Foundation of China (62572449). This work of Benyou Wang is supported by NSFC grant 72495131, the Shenzhen Science and Technology Program (JCYJ20220818103001002), Shenzhen Doctoral Startup Funding (RCBS20221008093330065), Shenzhen Science and Technology Program (Shenzhen Key Laboratory Grant No. ZDSYS20230626091302006), and Shenzhen Stability Science Program 2023. The work of Tian Ding is supported by Hetao Shenzhen-Hong Kong Science and Technology Innovation Cooperation Zone Project (No.HZQSWS-KCCYB-2024016). The work of Ruoyu Sun is supported by NSFC (No. 12326608); Hetao Shenzhen-Hong Kong Science and Technology Innovation Cooperation Zone Project (No.HZQSWS-KCCYB-2024016); University Development Fund UDF01001491, the Chinese University of Hong Kong, Shenzhen; Guangdong Provincial Key Laboratory of Mathematical Foundations for Artificial Intelligence (2023B1212010001).

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

# A  Experiment Implementation

## A.1  Training Implementation

For our experiments, we implemented several model variants with different training stages and architectures:

**32B Models:**

- **Prompt-Hint-SFT-32B**: Starting from the DeepSeek-R1-32B base model, we fine-tuned using 800 data samples with a learning rate of $1 \times 10^{-5}$, running for 17 epochs with a batch size of 96.

- **Hint-Engineering-SFT-32B**: Based on DeepSeek-R1-32B, we fine-tuned using only 30 high-quality, human-annotated data samples with a batch size of 96, learning rate of $1 \times 10^{-5}$, and 40 epochs.

- **Hint-Engineering-RFT-32B**: Building upon Hint-Engineering-SFT-32B, we further fine-tuned using 800 filtered data samples with a learning rate of $1 \times 10^{-5}$, 17 epochs, and batch size of 96.

**1.5B Models:**

- We distilled both Prompt-Hint-SFT-32B and Hint-Engineering-RFT-32B down to the DeepSeek-R1-1.5B architecture using 10k data samples with a learning rate of $7 \times 10^{-6}$, 6 epochs, and batch size of 128.

- For reinforcement learning, we adapted the veRL framework [62] to implement our specialized design outlined in Section 2.4. We further trained these 1.5B models with a learning rate of $1 \times 10^{-6}$, maximum response length of 16,000 tokens, 8 rollouts per problem, and maximum function calls limited to 15 per response, with each function call having a maximum length of 16,000 tokens.

- The RL training data was carefully selected by computing the average accuracy over 8 samples (avg@8) on 20k randomly selected problems from the NuminaMath-1.5 [63] dataset, then selecting only 1k challenging problems where avg@8 = 1/8 for focused training. This selective approach is motivated by our data ablation studies (Appendix D.2), which demonstrate that training on hard queries, while requiring longer convergence time, ultimately yields superior performance compared to easy or uniformly distributed queries.

## A.2  Evaluation Methodology

To ensure comprehensive and fair comparisons across different approaches, we implemented the following evaluation protocol:

- **Fair Comparison**: For publicly available models, we re-evaluated them on our local infrastructure using their original evaluation scripts to ensure consistent comparison conditions across all models.

- **Evaluation Protocol**: For all datasets, we extract the final answer from each model response and compare it directly to the ground truth using Math-Verify [64], considering a problem correctly solved only when Math-Verify returns True.

- **Evaluation Metrics**: We primarily used pass@1 as our base metric. Concretely, we employed avg@16 (average accuracy over 16 samples) as pass@1 for AIME24, AIME25, and AMC23 datasets. For MATH500 and OlympiadBench datasets, we used avg@4 as pass@1 due to their significantly larger test sizes.

- **Inference Setting**: Across all evaluations, we standardized inference parameters with maximum sequence length of 32,768 tokens, maximum function calls limited to 15, maximum tokens per function call set to 32,768, temperature of 0.6, and top-$p$ sampling parameter of 0.95.

### A.3 Function Call Limiting Strategy

During both evaluation and reinforcement learning rollout sampling with Python usage, we implemented a mechanism to handle scenarios where models reached the maximum allowed function calls. When this limit was reached, we appended the following system message to guide further reasoning:

```
[SYSTEM]
You have exceeded the allowed number of code executions. You can no longer
write or run code. Please continue solving the problem using your reasoning
and analytical skills.
```

This approach ensures that models can still complete their reasoning process without unlimited computational resources, better reflecting real-world usage constraints.

### A.4 Computational Hardware

Our experiments utilized the following hardware:

- **Training**: All training procedures, including supervised fine-tuning (SFT), rejection fine-tuning (RFT), and reinforcement learning (RL), were conducted on 4 servers, each equipped with 8 NVIDIA A100 GPUs.
- **Evaluation**: All model evaluations were performed on single servers, each equipped with 8 NVIDIA A100 GPUs, ensuring consistent measurement conditions across all compared approaches.

## B   Hint-engineering Process

**The Iterative Hint-Engineering Loop**   Hint-engineering implements an iterative refinement procedure that converts imperfect reasoning trajectories into expert-aligned ones. For a problem instance $P$, let $\tau^{(i)}$ denote the trajectory produced at iteration $i$. The loop operates as follows:

1. **Initial generation ($i = 0$).** The reasoner produces an initial trajectory $\tau^{(0)}$ for $P$.
2. **Annotation and evaluation.** A human annotator reviews $\tau^{(i)}$. If no deviation from the desired reasoning is detected, the procedure terminates with the final trajectory $\tau^{=\tau^{(i)}}$. Otherwise, the annotator localizes the erroneous step $t$ and its associated action $a_t$, and formulates a corrective hint $h_i$.
3. **Localized revision and resumption.** The context at step $t$ is augmented with $h_i$, yielding an updated state. From this state, the reasoner resumes its computation and produces the refined trajectory $\tau^{(i+1)}$.

The process iterates until $\tau^{(i)}$ meets the acceptance criteria or the allotted budget is exhausted.

## C   Evaluation Dataset Description

To comprehensively assess the mathematical reasoning capabilities of our models, we utilize several challenging benchmarks that span diverse difficulty levels and mathematical domains:

**AIME24 and AIME25.**   The American Invitational Mathematics Examination (AIME) represents a significant advancement beyond standard high school mathematics competitions, featuring problems that demand sophisticated reasoning techniques. We employ AIME24 and AIME25 as our primary benchmarks for evaluating advanced mathematical reasoning capabilities in our models.

**AMC23.**   Following DeepScaleR [65], we utilize their American Mathematics Competition (AMC) test set, which presents problems of moderate yet substantial difficulty, requiring considerable mathematical insight to solve. This dataset enables us to evaluate our models' proficiency in addressing a wider spectrum of mathematical challenges typically encountered in standard high school competitions.

**MATH500.** Curated from the test split of OpenAI's PRM800K dataset [24], MATH500 encompasses 500 carefully selected problems that represent a diverse range of mathematical challenges. We utilize this dataset to evaluate our models' ability to generalize across varied mathematical topics and problem structures.

**OlympiadBench.** Following DeepScaleR [65], we incorporate their OlympiadBench [66] into our evaluation framework. This benchmark comprises 675 Olympiad-level problems sourced from elite mathematics competitions, providing an exceptionally rigorous test of advanced mathematical reasoning.

# D Extra Experiments

## D.1 Code Behavior Evolution During RL

Figure 8: Evolution of code behavior metrics during RL training on AIME24.

Figure 8 tracks six key metrics of code behavior throughout the RL training process:

- **Code Usage Rate**: The percentage of responses containing Python code out of all responses. Both approaches show increasing code usage rates during training, with Hint-Engineering consistently maintaining higher rates, starting at 86% and quickly rising above 95%.

- **Code Success Rate**: The percentage of code blocks that execute without errors. The success rate shows different patterns between the two approaches. Prompt-Hint maintains relatively stable success rates around 78%, while Hint-Engineering shows more variability but achieves higher peaks.

- **Average Code Blocks**: The average number of Python code blocks per response. Interestingly, Hint-Engineering shows a steady decrease in the average number of code blocks from 4.4 to around 3.5 at peak performance, while Prompt-Hint increases from 1.9 to around 2.7. This divergence reveals a fundamental difference in evolution: Hint-Engineering evolves toward more efficient code usage (fewer but more effective blocks), while Prompt-Hint develops more code integration capabilities from its lower starting point.

These patterns reveal that reinforcement learning not only improves raw performance but actively shapes code usage behavior, with Hint-Engineering evolving toward an efficiency-optimized pattern (less code but more effective) while Prompt-Hint evolves toward increased code integration (more code with improving effectiveness).

## D.2 RL Training Data Ablation

To understand the impact of training data characteristics on RL performance, we conduct three ablation studies examining data volume, query difficulty distribution, and topic distribution effects.

**Data Scaling Ablation** (Figure 9, left): We investigate whether increasing training data volume directly improves optimal performance by comparing 1K, 5K, and 20K training samples while maintaining uniform difficulty and topic distributions. Surprisingly, we find that simply scaling up RL training data does not lead to better optimal performance. This finding suggests that the "less is

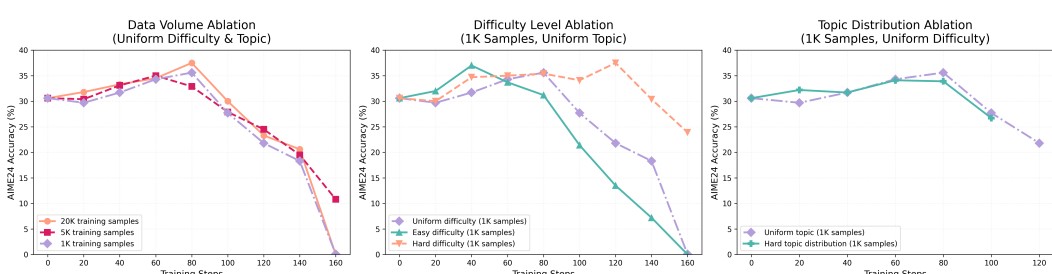

Figure 9: RL training data ablation studies on AIME24. **Left:** Data scaling ablation with 1K, 5K, and 20K training samples under uniform difficulty and topic distributions. **Middle:** Query difficulty ablation comparing easy (avg@8=7/8), uniform, and hard (avg@8=1/8) difficulty distributions with 1K samples. **Right:** Topic distribution ablation comparing uniform and hard topic distributions with 1K samples under uniform difficulty.

more" principle still holds in large reasoning model (LRM) training, and data quality may be more important than quantity for RL fine-tuning.

**Query Difficulty Ablation** (Figure 9, middle): We examine how query difficulty distribution affects learning dynamics by comparing three settings with 1K samples: easy queries (avg@8=7/8), uniform difficulty distribution, and hard queries (avg@8=1/8). Our results reveal distinct learning patterns: easy queries achieve optimal performance earliest but with lower peak accuracy, uniform distribution shows intermediate behavior, while hard queries take longer to reach optimal performance but ultimately achieve the best results. This suggests that training on challenging examples, though slower to converge, leads to superior final performance in mathematical reasoning tasks.

**Topic Distribution Ablation** (Figure 9, right): We investigate whether aligning training topic distribution with hard query topics improves performance by comparing uniform topic distribution against a distribution matching hard queries (avg@8=1/8). The results show minimal differences between the two distributions, indicating that topic distribution changes do not significantly impact optimal performance. This suggests that the model's reasoning capabilities generalize well across different mathematical topics, and the difficulty level is more crucial than specific topic coverage.

### D.3 Code Reward Ablation

To investigate the effectiveness of our code reward mechanism and determine the optimal penalty strength, we conduct ablation studies comparing different penalty values in the code reward function.

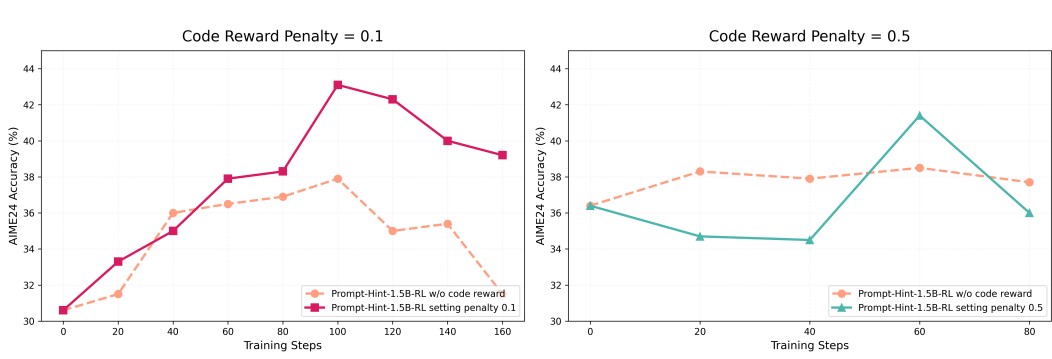

Figure 10: Code reward penalty ablation study on AIME24. **Left:** Comparison between training with and without code reward using penalty=0.1. **Right:** Comparison between training with and without code reward using penalty=0.5 on a different SFT model.

**Code Reward Effectiveness**: Our results demonstrate that incorporating code reward significantly improves model performance compared to training without it. In the penalty=0.1 setting (Figure 10, left), the model with code reward achieves a peak accuracy of 43.1% at step 100, substantially outperforming the baseline without code reward (37.9% peak). This improvement persists throughout most of the training process, indicating that the code reward provides consistent learning signals that guide the model toward better reasoning strategies.

**Penalty Strength Analysis**: Comparing different penalty values reveals that penalty=0.1 yields superior and more stable performance than penalty=0.5. The penalty=0.1 configuration shows smoother convergence and maintains higher accuracy levels across training steps. In contrast, penalty=0.5 (Figure 10, right) exhibits more erratic behavior, with a notable spike at step 60 (41.4%) followed by a sharp decline. This suggests that excessive penalty strength may introduce instability in the training process, potentially causing the model to over-correct its behavior when generating incorrect code.

### D.4 Token Efficiency Analysis for Lightweight Models

We conduct a detailed analysis of token efficiency in 1.5B parameter lightweight models, examining how different approaches impact computational resource utilization while maintaining mathematical reasoning capabilities.

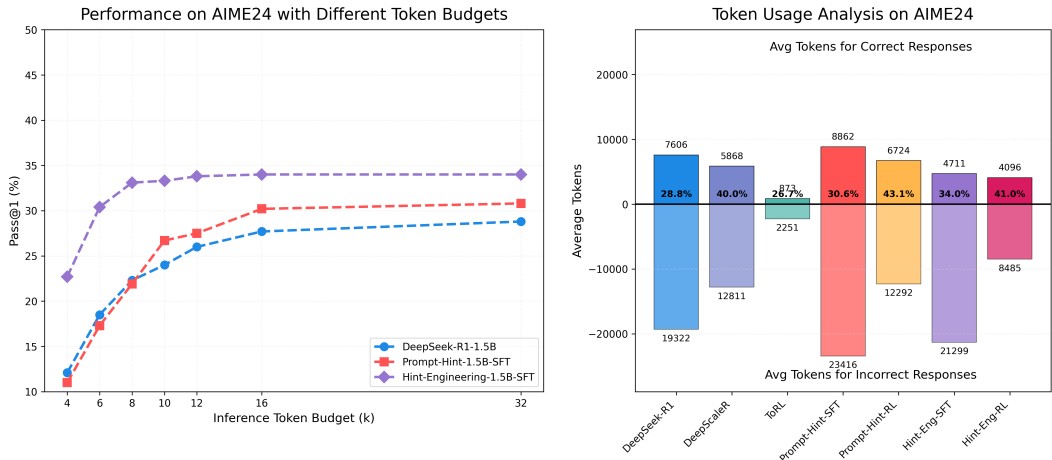

Figure 11: Token efficiency analysis for 1.5B parameter models on AIME24. **Left:** Performance comparison across different token budgets. **Right:** Detailed token usage breakdown for each model, showing average token consumption for both correct (above axis) and incorrect (below axis) responses, with Pass@1 accuracy displayed at the center.

Our token efficiency analysis reveals two key insights about the computational efficiency of lightweight mathematical reasoning models:

**Superior Performance with Limited Token Budgets:** As shown in Figure 11 (left), Hint-Engineering-SFT consistently outperforms both the base model (DeepSeek-R1-1.5B) and Prompt-Hint-SFT across all token budget constraints. Most notably, with just a 4k token budget, Hint-Engineering-SFT achieves 22.7% accuracy, nearly double the performance of DeepSeek-R1-1.5B (12.1%) and Prompt-Hint-SFT (11.0%). This indicates that Hint-Engineering's structured approach to problem-solving enables more efficient reasoning within constrained computational environments.

**Substantial Token Savings Across All Response Types:** Figure 11 (right) demonstrates that Hint-Engineering models maintain significantly lower token usage compared to alternatives. For correct responses, Hint-Engineering-SFT uses 47% fewer tokens than Prompt-Hint-SFT (4,711 vs. 8,862), while for incorrect responses, the savings are even more dramatic, with Hint-Engineering-RL consuming 31% fewer tokens than Prompt-Hint-RL (8,485 vs. 12,292). Overall, Hint-Engineering-RL achieves a 32% reduction in total token consumption compared to Prompt-Hint-RL (6,684 vs.

9,891) while maintaining comparable accuracy (41.0% vs. 43.1%). Furthermore, Hint-Engineering models use about 50% fewer tokens for the 1.5B model compared with the natural language models.

## D.5 Pass@K Analysis for Frontier Models

To understand how our approaches scale with sampling budget and model size, we conduct a comprehensive Pass@k analysis on 32B parameter frontier models.

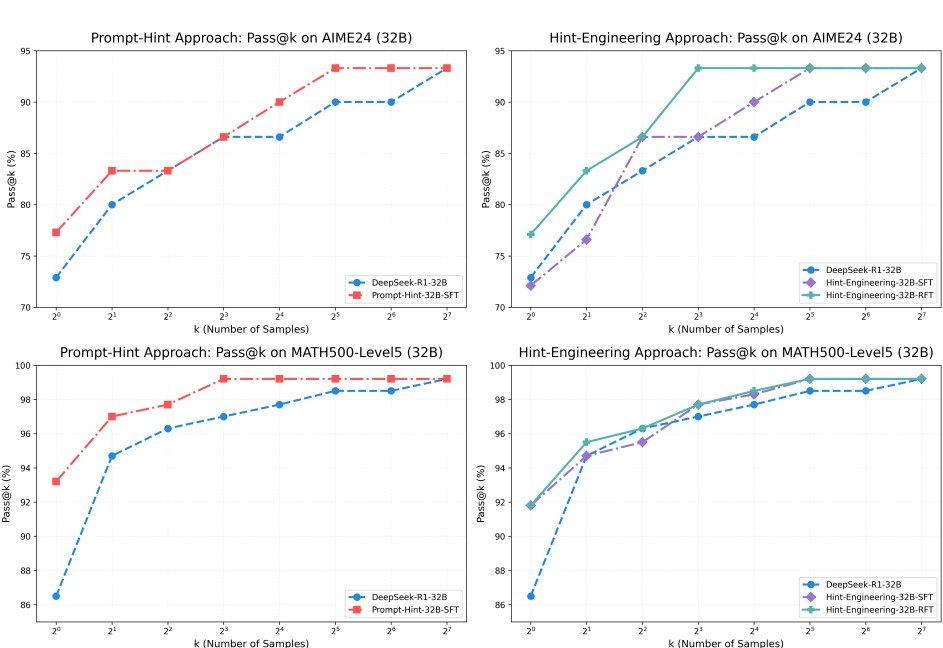

Figure 12: Pass@k analysis for 32B parameter models. **Top row:** Performance on AIME24 comparing Prompt-Hint (left) and Hint-Engineering (right) approaches against the DeepSeek-R1-32B baseline. **Bottom row:** Performance on MATH500-Level5 showing similar patterns.

Figure 12 illustrates the performance of our 32B parameter models as a function of sample size ($k$) on both AIME24 and MATH500-Level5 datasets. Our analysis reveals two key patterns in the scaling behavior of these models.

**SFT Provides Modest Gains at Lower Sampling Budgets:** Both Prompt-Hint-32B-SFT and Hint-Engineering-32B-SFT show improvements over the baseline DeepSeek-R1-32B, particularly at lower k values. At k=1 on AIME24, Prompt-Hint-32B-SFT achieves 77.3% accuracy compared to the baseline's 72.9%. However, all models eventually converge to similar maximum performance levels (93.3% for AIME24 and 99.2% for MATH500-Level5) as k increases, suggesting that SFT primarily enhances the model's efficiency rather than raising its reasoning ceiling.

**RFT Significantly Enhances Sample Efficiency:** Hint-Engineering-32B-RFT demonstrates remarkable efficiency, reaching 93.3% accuracy with just k=8 samples on AIME24, while other approaches require 2-4 times more samples to achieve comparable results. This indicates that reinforcement fine-tuning successfully optimizes the model's ability to consistently arrive at correct solutions with fewer attempts, making it particularly valuable in scenarios where computational efficiency is critical.

## D.6 Statistical Significance Testing

To rigorously validate our empirical results and ensure that the observed improvements are statistically meaningful, we conducted pairwise Wilcoxon signed-rank tests. This non-parametric statistical test is well-suited for comparing the performance of two models on a per-problem basis, without assuming a normal distribution of performance differences.

We followed the procedure outlined in our rebuttal: we pooled the results across all problems from the five mathematical benchmarks to create a sufficiently large sample for robust analysis. The test was performed separately for our 32B frontier models and 1.5B lightweight models to assess statistical significance at different scales. We evaluated two key dimensions: accuracy improvements and token reduction.

The results of our statistical tests are presented in Table 2.

Table 2: Statistical significance of performance gains, evaluated using pairwise Wilcoxon signed-rank tests on pooled results from all benchmarks. The p-values confirm that our efficiency gains are highly significant, and accuracy improvements are either significant or show a strong positive trend.

| Model Scale | Model A (Ours) | Model B (Baseline) | Acc. Improv. | p-value (Acc.) | Token Reduction | p-value (Tokens) |
|---|---|---|---|---|---|---|
| 32B | Hint-Eng-RFT | R1-distill-32B | +3.80% | 0.055 | 36.3% | < 0.001 |
| 1.5B | Hint-Eng-RL | R1-distill-1.5B | +7.41% | 0.013 | 59.1% | < 0.001 |

The analysis confirms the following:

- **Token Efficiency Gains are Highly Significant**: For both the 32B and 1.5B model scales, the reduction in token consumption is statistically highly significant, with $p < 0.001$. This provides strong evidence that the CoRT framework induces a fundamentally more efficient reasoning process.
- **Accuracy Improvements are Statistically Meaningful**: For the 1.5B model, the accuracy improvement of +7.41% is statistically significant ($p = 0.013$). For the 32B model, the +3.80% improvement shows a strong positive trend that approaches statistical significance ($p = 0.055$).

These results, combined with the fact that our reported metrics in Table 1 are averages over multiple stochastic samples (16 for AIME/AMC and 4 for MATH/Olympiad), robustly support our claim that the CoRT framework leads to meaningful and reliable improvements in both reasoning accuracy and efficiency.

# E Prompts

---

**Hint-Prompt and Hint-Engineering Prompt**

Given a mathematical problem, follow the instructions below to solve it.
Instructions:
When solving mathematical problems, you should leverage both natural language reasoning and interactive Python code execution. Your goal is to provide clear, detailed explanations while utilizing Python to perform complex calculations, symbolic manipulations, data analysis, or any other tasks that can aid in problem-solving. Follow these guidelines to ensure a coherent and effective response:

1. **Natural Language Reasoning:**
- Provide comprehensive, step-by-step explanations of your thought process.
- Ensure that each step logically follows from the previous one, maintaining clarity and coherence.
- Use appropriate mathematical terminology and notation where necessary.
- Planning, Modeling, and Analysis:

- Use natural language to outline the overall approach to the problem.
- Develop mathematical models or representations as needed.
- Analyze the problem to determine the best strategies for finding a solution.

2. **Inserting Python Code Blocks:**
- When a Python code snippet can aid in analysis, computation, or symbolic manipulation, insert a Python code block.
- Use triple backticks with 'python' to denote the start of a Python code block and triple backticks to close it.

---

- Example:
'''python
'''
3. **Displaying Code Output:**
- Immediately after a Python code block, present the output generated by the code.
- Use triple backticks with 'output' to denote the start of the output block and triple backticks to close it.
- Example:
'''output
'''
4. **Encouraging Multiple Python Calls and Diverse Functionality:**
- Utilize Python multiple times throughout your solution to handle different aspects of the problem.
- Take advantage of various Python libraries and functionalities such as:
- 'numpy' for numerical computations
- 'scipy' for scientific computing and advanced mathematical functions
- 'sympy' for symbolic mathematics
- 'pandas' for data manipulation and analysis
- 'math' for fundamental mathematical operations
- 'statistics' for statistical computations
- 'fractions' for rational number calculations
- Ensure that each Python snippet is purposeful and enhances the understanding or resolution of the problem.
- **Specific Calculations and Complex Operations:**
- Use Python to perform detailed calculations that would be cumbersome by hand.
- Implement complex algorithms or data processing tasks that facilitate the solution.
- Handle any intricate operations that support the overall analysis and modeling of the problem

Problem:

---

## Code Behavior Analysis Prompt

You are an expert in analyzing code and understanding its purpose, especially within the context of mathematical problem-solving. Your task is to analyze Python code snippets within a solution to a mathematical problem and classify each snippet based on its purpose.

You will be given a problem/solution pair. The solution may contain multiple Python code snippets. For each Python code snippet, you must determine:

1. **Is it Verification or Calculation?**
- **Verification:** The Python code *verifies* a result or conclusion that was already reached through reasoning in the solution. The Python code confirms a pre-existing answer or property.
- **Calculation:** The Python code *calculates* a result that was NOT explicitly present in the solution's reasoning up to that point. The Python code derives a new, previously unknown answer or intermediate value.

2. **What is the specific function of the Python code snippet?** Choose one or more from the following list of functions (be as specific as possible). If none of these functions are appropriate, provide a brief (one sentence) description of the function of the code.

1. **Solving Equations and Systems of Equations**
- Finding numerical or symbolic solutions to algebraic, differential, and other types of equations.

2. **Symbolic Mathematics and Manipulation**
- Performing algebraic operations such as differentiation, integration, simplification, and expansion using symbolic math libraries like SymPy.

3. **Numerical Approximation Methods**
- Approximating solutions for problems that lack analytical solutions, including numerical integration, root finding, and solving differential equations.

4. **Data Visualization and Plotting**
- Creating graphs, charts, and other visual representations using libraries like Matplotlib and Seaborn to illustrate mathematical concepts and data patterns.

5. **Pattern Recognition and Analysis**
- Identifying and analyzing patterns or relationships in data using statistical and machine learning techniques.

6. **Optimization and Solution Searching**
- Implementing algorithms to find optimal solutions to problems, including linear programming, integer programming, and heuristic methods.

7. **Property Verification and Theorem Checking**
- Verifying mathematical properties and theorems for given inputs using computational methods.

8. **Modular Arithmetic and Number Theory Operations**
- Performing calculations involving modular arithmetic, such as finding inverses, solving congruences, and applying the Chinese Remainder Theorem.

9. **Prime Number Testing and Factorization**
- Determining the primality of numbers and performing prime factorization using efficient algorithms.

10. **Geometric and Computational Geometry Calculations**
- Calculating areas, volumes, distances, angles, convex hulls, intersections, and performing geometric transformations.

11. **Probability, Statistics, and Simulations**
- Computing probabilities, expected values, variances, and running Monte Carlo simulations to model random processes.

12. **Linear Algebra: Matrix and Vector Operations**
- Performing matrix multiplication, inversion, eigenvalue decomposition, and other linear algebra operations using libraries like NumPy and SciPy.

13. **Data Generation and Simulation**
- Creating synthetic data sets and simulating mathematical models to explore and analyze behaviors.

14. **Combinatorial Enumeration and Game Theory**
- Counting permutations, combinations, and analyzing combinatorial games to determine winning strategies.

15. **Graph Theory Algorithms**
- Implementing algorithms for graph traversal (DFS, BFS), shortest paths (Dijkstra's, Floyd-Warshall), and finding minimum spanning trees (Kruskal's, Prim's).

16. **Dynamic Programming and Recurrence Relations**
- Designing dynamic programming solutions and solving linear and non-linear recurrence relations to find closed-form expressions.

17. **Fast Fourier Transforms (FFT) and Signal Processing**
- Utilizing FFT for problems involving polynomial multiplication, number-theoretic transforms, and analyzing frequency components.

18. **Boolean Algebra and Logic Operations**
- Manipulating and simplifying logical expressions, constructing truth tables, and solving Boolean equations.

19. **Big Integer and Arbitrary-Precision Arithmetic**
- Handling calculations with very large integers beyond standard data type limits using Python's arbitrary-precision capabilities.

20. **Symbolic Integration, Differentiation, and Proof Verification**
- Performing advanced calculus operations and assisting in verifying mathematical proofs using symbolic computation libraries.

21. **Linear Programming and Optimization Techniques**
- Formulating and solving linear optimization problems using libraries like PuLP and SciPy.

22. **Algorithm Optimization and Numerical Stability**
- Enhancing algorithm performance by improving time and space complexity and ensuring numerical stability for accurate results.

23. **Automated Theorem Proving and Symbolic Logic**
- Utilizing tools and libraries to automatically prove mathematical theorems and manipulate symbolic logic statements.

24. **Data Structures Implementation and Management**
- Creating and utilizing advanced data structures such as trees, graphs, and heaps to efficiently solve competition problems.

25. **Fractal and Recursive Pattern Generation**
- Creating and analyzing fractals and other recursive patterns relevant to geometry and combinatorics problems.

You will be given a "Problem" and a "Solution". The solution may contain one or more "Python code" snippets. For *each* Python code snippet, provide the following output:

Python Code idx: [The idx of Python code](for example 1,2,3)
Classification: [Verification or Calculation]
Function: [one or more specific functions from the list above OR some one-sentence descriptions]

Problem:

Solution:

## F  Case Study

### F.1  5 Samples of $D_{Hint-engineering-SFT}$

**Sample 1**

*Problem*:
Find the least positive integer $n$ for which $2^n + 5^n - n$ is a multiple of 1000.
*Solution*:
Let's solve this step by step. The problem is to find the least positive integer n such that
$2^n + 5^n - n \equiv 0 \pmod{1000}$.
Since $1000 = 8 \times 125$, by the Chinese Remainder Theorem, this is equivalent to solving:

$$2^n + 5^n - n \equiv 0 \pmod{8}$$
$$2^n + 5^n - n \equiv 0 \pmod{125}$$

For modulo 8: - When $n \geq 3$: $2^n \equiv 0 \pmod{8}$ - $5^n \pmod{8}$ cycles every 2: $5, 1, 5, 1, ...$ - After analysis, we find $n \equiv 5 \pmod{8}$
For modulo 125, let's first find the period of $2^n \bmod 125$:

**Python Code 1**

```python
def find_period_mod125():
    seen_values = {}
    current_value = 1
    position = 0
    sequence = []

    while True:
        if current_value in seen_values:
            period_length = position - seen_values[current_value]
            period_sequence = sequence[seen_values[current_value]...
            return period_length, period_sequence

        seen_values[current_value] = position
        sequence.append(current_value)
        current_value = (current_value * 2) % 125
        position += 1

period_length, period_sequence = find_period_mod125()
print(f"Period length: {period_length}")

print("\nExample cases:")
test_numbers = [0, 1, 2, period_length-1]
for n in test_numbers:
    result = pow(2, n, 125)
    print(f"2^{n} mod 125 = {result}")
```

**Output 1**

```
Period length: 100

Example cases:
2^0 mod 125 = 1
2^1 mod 125 = 2
2^2 mod 125 = 4
2^99 mod 125 = 63
```

Similarly, let's check the period of $6^k \bmod 125$:

```
Python Code 2

def find_period_mod125():
    seen_values = {}
    current_value = 1
    position = 0
    sequence = []

    while True:
        if current_value in seen_values:
            period_length = position - seen_values[current_value]
            period_sequence = sequence[seen_values[current_value]...
            return period_length, period_sequence

        seen_values[current_value] = position
        sequence.append(current_value)
        current_value = (current_value * 6) % 125
        position += 1

period_length, period_sequence = find_period_mod125()
print(f"Period length: {period_length}")

print("\nExample cases:")
test_numbers = [0, 1, 2, period_length-1]
for n in test_numbers:
    result = pow(6, n, 125)
    print(f"6^{n} mod 125 = {result}")
```

```
Output 2

Period length: 25

Example cases:
6^0 mod 125 = 1
6^1 mod 125 = 6
6^2 mod 125 = 36
6^24 mod 125 = 21
```

[...detailed analysis omitted...]

After analysis, we need to solve:

$$32 \cdot 6^k \equiv 8k + 5 \pmod{125}$$

Let's find the solution:

```
Python Code 3

def find_k():
    for k in range(0, 150):
        mod_6k = pow(6, k, 125)
        lhs = (32 * mod_6k) % 125
        rhs = (8*k + 5) % 125
        if lhs == rhs:
            print(f"Found k={k}, n=8*{k}+5={8*k+5}")
            return 8*k+5
    return None

result = find_k()
print(result)
```

```
Output 3

Found k=99, n=8*99+5=797
797
```

Therefore, $k = 99$ gives us $n = 8 \times 99 + 5 = 797$.
Verification: - $5^{797} \equiv 125 \pmod{1000}$ (since $5^n$ cycles 125, 625 for odd/even $n \geq 3$) - $2^{797} \equiv 672 \pmod{1000}$ - Therefore, $2^{797} + 5^{797} - 797 \equiv 0 \pmod{1000}$
The answer is $\boxed{797}$.

---

## Sample 2

***Problem***:
Find the largest prime number $p < 1000$ for which there exists a complex number $z$ satisfying: (1) the real and imaginary part of $z$ are both integers; (2) $|z| = \sqrt{p}$, and (3) there exists a triangle whose three side lengths are $p$, the real part of $z^3$, and the imaginary part of $z^3$.
***Solution***:
Let's try to solve this problem step by step. The goal is to find the largest prime number $p < 1000$ such that there exists a complex number $z$ with integer real and imaginary parts, $|z| = \sqrt{p}$, and there's a triangle with side lengths $p$, $\text{Re}(z^3)$, and $\text{Im}(z^3)$.
First, let's understand the problem. We need to find a prime $p < 1000$. The complex number $z$ must have integer real and imaginary parts, so $z = a + bi$ where $a$ and $b$ are integers. The magnitude $|z|$ is $\sqrt{a^2 + b^2} = \sqrt{p}$, which implies that $a^2 + b^2 = p$. Therefore, $p$ must be a prime that can be expressed as the sum of two squares. That's a key point.
[...detailed analysis omitted...]
So the algorithm we will implement is:
For each prime $p$ (starting from largest to smallest):

- Find $(a, b)$ using find_a_b$(p)$

- Generate all variations:
  $[(a, b), (-a, b), (a, -b), (-a, -b), (b, a), (-b, a), (b, -a), (-b, -a)]$

- For each variation $(x, y)$:
  - Compute $\text{Re} = x^3 - 3xy^2$
  - Compute $\text{Im} = 3x^2y - y^3$
  - If $\text{Re} > 0$ and $\text{Im} > 0$:
    * Check triangle inequalities:
      · $p + \text{Re} > \text{Im}$
      · $p + \text{Im} > \text{Re}$
      · $\text{Re} + \text{Im} > p$

* If all three are true, then return $p$ as a valid solution

Let's implement this in Python:

```python
import math

def sieve(n):
    sieve = [True] * (n + 1)
    sieve[0] = sieve[1] = False
    for i in range(2, int(math.sqrt(n)) + 1):
        if sieve[i]:
            sieve[i * i : n + 1 : i] = ...
    primes = [i for i, is_prime in enumerate(sieve) if is_prime]
    return primes

def find_a_b(p):
    max_a = int(math.isqrt(p))
    for a in range(max_a + 1):
        b_sq = p - a * a
        if b_sq < 0:
            continue
        b = int(math.isqrt(b_sq))
        if b * b == b_sq:
            return (a, b)
    return None

def is_triangle(p, re, im):
    return (re + im > p) and (p + re > im) and (p + im > re)

def check_primes_for_triangle():
    eligible_primes = ...
    eligible_primes.sort(reverse=True)

    for p in eligible_primes:
        ab = find_a_b(p)
        if ab is None:
            continue
        a, b = ab
        variations = [
            (a, b), (-a, b), (a, -b), (-a, -b),
            (b, a), (-b, a), (b, -a), (-b, -a)
        ]

        for x, y in variations:
            re = x ** 3 - 3 * x * y ** 2
            im = 3 * x ** 2 * y - y ** 3
            if re > 0 and im > 0:
                if is_triangle(p, re, im):
                    print(f"Found valid prime ...")
                    print(f"Checking triangle inequalities: ...")
                    print(f"Answer: {p}")
                    return

check_primes_for_triangle()
```

Therefore, the largest prime that satisfies all conditions is $\boxed{349}$.

## Sample 3

***Problem***:

Each face of two noncongruent parallelepipeds is a rhombus whose diagonals have lengths $\sqrt{21}$ and $\sqrt{31}$. The ratio of the volume of the larger of the two polyhedra to the volume of the smaller is $\frac{m}{n}$, where $m$ and $n$ are relatively prime positive integers. Find $m + n$. A parallelepiped is a solid with six parallelogram faces such as the one shown below.

***Solution***:

Let's try to solve this problem step by step. The problem is about two noncongruent parallelepipeds where each face is a rhombus with diagonals of lengths $\sqrt{21}$ and $\sqrt{31}$. We need to find the ratio of their volumes (larger to smaller) and then compute $m + n$ where the ratio is $\frac{m}{n}$ in simplest terms.

[...detailed analysis omitted...]

Therefore, the two possible rhombohedrons would have all angles equal to $\arccos(5/26)$ or $\arccos(-5/26)$. Thus, these are two distinct rhombohedrons with the same edge length but different angles, hence noncongruent.

Let's compute the volumes using Python:

Python Code

```python
import sympy as sp

# Define the square root of 13 symbolically
s = sp.sqrt(13)

# Define angles symbolically using arccos
theta = sp.acos(sp.Rational(5, 26))
theta_prime = sp.acos(sp.Rational(-5, 26))

# Define cos(theta) and cos(theta_prime) as symbolic fractions
cos_theta = sp.Rational(5, 26)
cos_theta_prime = sp.Rational(-5, 26)

# Compute volume for theta using the symbolic expression
vol1_expr = s**3 * sp.sqrt(1 - 3*cos_theta**2 + 2*cos_theta**3)
vol1 = sp.simplify(vol1_expr)

# Compute volume for theta_prime using the symbolic expression
vol2_expr = s**3 * sp.sqrt(1 - 3*cos_theta_prime**2 + ...
vol2 = sp.simplify(vol2_expr)

# Print the symbolic results
print("Volume 1:", vol1)
print("Volume 2:", vol2)
```

The two volumes are $\frac{63\sqrt{2}}{2}$ and $31\sqrt{2}$. Since $\frac{63}{2} = 31.5 > 31$, $\frac{63\sqrt{2}}{2}$ is larger than $31\sqrt{2}$. Therefore, $\frac{m}{n} = \frac{63}{62}$. Since $63 = 7 \times 9$ and $62 = 2 \times 31$, they are coprime. So $m = 63$, $n = 62$, and $m + n = 125$.
Therefore, the answer is $\boxed{125}$.

## Sample 4

***Problem:***
Let $\triangle ABC$ be an equilateral triangle with side length 55. Points $D$, $E$, and $F$ lie on $\overline{BC}$, $\overline{CA}$, and $\overline{AB}$, respectively, with $BD = 7$, $CE = 30$, and $AF = 40$. Point $P$ inside $\triangle ABC$ has the property that
$$\angle AEP = \angle BFP = \angle CDP.$$
Find $\tan^2(\angle AEP)$.

***Solution:***
Let's try to solve this problem step by step. We have an equilateral triangle ABC with side length 55. Points D, E, F are on BC, CA, AB respectively, with BD=7, CE=30, AF=40. There's a point P inside the triangle such that the angles $\angle AEP$, $\angle BFP$, and $\angle CDP$ are equal. We need to find the square of the tangent of this angle.
First, let's place the triangle ABC in a coordinate system with $A(0,0)$, $B(55,0)$, and $C(\frac{55}{2}, \frac{55\sqrt{3}}{2})$. Then we can calculate the coordinates of points D, E, and F:
- Point D: $D(\frac{103}{2}, \frac{7\sqrt{3}}{2})$ - Point E: $E(\frac{25}{2}, \frac{25\sqrt{3}}{2})$ - Point F: $F(40, 0)$
[...detailed analysis omitted...]
For each angle $\theta$, we can write equations involving the slopes of the lines:
1. For $\angle AEP$:
$$\tan(\theta) = \frac{\frac{y - \frac{25\sqrt{3}}{2}}{x - \frac{25}{2}} - \sqrt{3}}{1 + \sqrt{3} \cdot \frac{y - \frac{25\sqrt{3}}{2}}{x - \frac{25}{2}}}$$

2. For $\angle BFP$:
$$\tan(\theta) = \frac{y}{x - 40}$$

3. For $\angle CDP$:
$$\tan(\theta) = \frac{\frac{y - \frac{7\sqrt{3}}{2}}{x - \frac{103}{2}} + \sqrt{3}}{1 - \sqrt{3} \cdot \frac{y - \frac{7\sqrt{3}}{2}}{x - \frac{103}{2}}}$$

Let's solve this system using Python:

```python
import sympy as sp

# Define variables
x, y = sp.symbols('x y')

# Coordinates of points
E_x = sp.Rational(25, 2)
E_y = sp.Rational(25, 2) * sp.sqrt(3)
F_x = sp.Rational(40, 1)
F_y = sp.Rational(0, 1)
D_x = sp.Rational(103, 2)
D_y = sp.Rational(7, 2) * sp.sqrt(3)

# Slopes and tan(theta) expressions
m_EA = sp.sqrt(3)
m_EP = (y - E_y) / (x - E_x)
tan_theta_E = ...

# Solve the system of equations
solution = sp.solve([eq1, eq2], (x, y))
print("Solutions for x and y:")
print(solution)
```

**Output**

```
Solutions for x and y:
[(1525/38, 25*sqrt(3)/38)]
```

With these coordinates, we can compute $\tan(\theta)$:

$\tan(\theta) = \frac{\frac{25\sqrt{3}}{38}}{\frac{1525}{38} - 40} = 5\sqrt{3}$

Therefore: $\tan^2(\theta) = (5\sqrt{3})^2 = 75$

Therefore, the answer is $\boxed{75}$.

---

**Sample 5**

***Problem***:
In $\triangle ABC$ with side lengths $AB = 13$, $BC = 14$, and $CA = 15$, let $M$ be the midpoint of $\overline{BC}$. Let $P$ be the point on the circumcircle of $\triangle ABC$ such that $M$ is on $\overline{AP}$. There exists a unique point $Q$ on segment $\overline{AM}$ such that $\angle PBQ = \angle PCQ$. Then $AQ$ can be written as $\frac{m}{\sqrt{n}}$, where $m$ and $n$ are relatively prime positive integers. Find $m + n$.

***Solution***:
Let's solve this step by step. First, let's establish a coordinate system: - Place B at (0,0) - Place C at (14,0) - Then M is at (7,0) - Using distances AB=13 and AC=15, we can find A at (5,12)
[...detailed analysis omitted...]
First, let's find the circumcircle of triangle ABC:

[...detailed analysis omitted...]

Next, let's find point P by finding where line AM intersects the circumcircle:

```python
from sympy import symbols, solve, Rational

# Define the variable
t = symbols('t')

# Define the parametric equations for x and y
x = 5 + 2*t
y = 12 - 12*t

# Define the equation of the circle
circle_eq = (x - 7)**2 + ...

# Solve for t
expanded_eq = circle_eq.expand()
solution_t = solve(expanded_eq, t)

# Calculate the coordinates for each solution t
points = []
for t_solution in solution_t:
    x_value = x.subs(t, t_solution)
    y_value = y.subs(t, t_solution)
    points.append((x_value, y_value))

print("Solutions for t:", solution_t)
for idx, point in enumerate(points):
    print(f"Coordinates of point for t={solution_t[idx]}...")
```

**Output 2**

```
Solutions for t: [0, 197/148]
Coordinates of point for t=0: (x, y) = (5, 12)
Coordinates of point for t=197/148: (x, y) = (567/74, -147/37)
```

[...detailed analysis omitted...]

Now, let's find point Q using the condition that $\angle PBQ = \angle PCQ$:

```python
from sympy import symbols, Eq, solve, sqrt, simplify, fraction
from sympy import Rational

t = symbols('t')

# Coordinates of Q
Qx = 5 + 2*t
Qy = 12 - 12*t

# Coordinates of P
Px = Rational(567, 74)
Py = Rational(-147, 37)

# Vectors BP, BQ, CP, CQ
BPx = Px - 0
BPy = Py - 0
BQx = Qx - 0
BQy = Qy - 0

CPx = Px - 14
CPy = Py - 0
CQx = Qx - 14
CQy = Qy - 0

# Compute dot products and magnitudes
dot_BP_BQ = BPx * BQx + BPy * BQy
dot_CP_CQ = CPx * CQx + CPy * CQy

magnitude_BP = sqrt(BPx**2 + BPy**2)
magnitude_BQ = sqrt(BQx**2 + BQy**2)
magnitude_CP = sqrt(CPx**2 + CPy**2)
magnitude_CQ = sqrt(CQx**2 + CQy**2)

# Set up and solve the equation
equation = Eq(dot_BP_BQ/(magnitude_BP * magnitude_BQ),
              dot_CP_CQ/(magnitude_CP * magnitude_CQ))
solution = solve(equation, t)

print("Solutions for t:", solution)
```

```
Solutions for t: [99/148, 197/148]
```

[...detailed analysis omitted...]

Finally, let's compute $AQ$ and verify our answer:

```python
from math import gcd

# First compute AQ distance
Qx_val = Qx.subs(t, Rational(99,148))
Qy_val = Qy.subs(t, Rational(99,148))
AQ_distance = sqrt((Qx_val - 5)**2 + (Qy_val - 12)**2)
simplified_AQ = AQ_distance.simplify()
print("AQ =", simplified_AQ)

# Verify m and n are coprime
m = 99
n = 148
gcd_val = gcd(m, n)
print("GCD of 99 and 148:", gcd_val)
```

Output 4

```
AQ = 99*sqrt(37)/74
GCD of 99 and 148: 1
```

After manipulation, we find $AQ = \frac{99}{\sqrt{148}}$ where $m = 99$ and $n = 148$ are coprime positive integers.

Therefore, $m + n = 99 + 148 = \boxed{247}$.

## F.2 Test case of Prompt-Hint-SFT-32B

**A case from AIME24**

***Problem:***
Let $\mathcal{B}$ be the set of rectangular boxes with surface area $54$ and volume $23$. Let $r$ be the radius of the smallest sphere that can contain each of the rectangular boxes that are elements of $\mathcal{B}$. The value of $r^2$ can be written as $\frac{p}{q}$, where $p$ and $q$ are relatively prime positive integers. Find $p + q$.

***Solution:***
The problem is about finding the smallest sphere that can contain any rectangular box from the set B, where each box has a surface area of 54 and volume 23. The radius of this minimal sphere would be half of the maximum space diagonal among all possible boxes.

For a box with side lengths $a$, $b$, $c$, we have: 1. Space diagonal $d = \sqrt{a^2 + b^2 + c^2}$ 2. Surface area: $2(ab + bc + ac) = 54$ 3. Volume: $abc = 23$

Therefore, $r^2 = \frac{a^2 + b^2 + c^2}{4}$, and we need to maximize $a^2 + b^2 + c^2$ subject to the constraints.
[...detailed analysis omitted...]

Let's approach this using Lagrange multipliers. Our Lagrangian is:
$$L = a^2 + b^2 + c^2 - \lambda(2(ab + bc + ac) - 54) - \mu(abc - 23)$$

Taking partial derivatives:
$$\frac{\partial L}{\partial a} = 2a - \lambda(2b + 2c) - \mu bc = 0$$
$$\frac{\partial L}{\partial b} = 2b - \lambda(2a + 2c) - \mu ac = 0$$
$$\frac{\partial L}{\partial c} = 2c - \lambda(2a + 2b) - \mu ab = 0$$

After analysis, we find that the maximum occurs when: - $a = b = 2$ - $c = \frac{23}{4}$
Let's verify this solution:

**Python Code**

```python
from sympy import symbols, Eq, solve, sqrt
from sympy import Rational

# Define variables
a = 2
b = 2
c = Rational(23, 4)

# Check surface area
surface_area = 2 * (a*b + b*c + a*c)
print(f"Surface area = {surface_area}")

# Check volume
volume = a * b * c
print(f"Volume = {volume}")

# Calculate space diagonal squared
diagonal_squared = a**2 + b**2 + c**2
print(f"Space diagonal squared = {diagonal_squared}")

# Calculate r squared
r_squared = diagonal_squared / 4
print(f"r squared = {r_squared}")
```

**Output**

```
Surface area = 54
Volume = 23
Space diagonal squared = 657/16
r squared = 657/64
```

Therefore: - Space diagonal squared = $\frac{657}{16}$ - $r^2 = \frac{657}{64}$
Since $657 = 3 \times 3 \times 73$ and $64 = 2^6$, these numbers are coprime. Therefore, $p = 657$ and $q = 64$.
The answer is $p + q = 657 + 64 = \boxed{721}$.

---

**A Case from AIME25**

***Problem***:
Six points $A, B, C, D, E$ and $F$ lie in a straight line in that order. Suppose that $G$ is a point not on the line and that $AC = 26$, $BD = 22$, $CE = 31$, $DF = 33$, $AF = 73$, $CG = 40$, and $DG = 30$. Find the area of $\triangle BGE$.
***Solution***:
Let's solve this step by step:
1) First, let's assign coordinates to the points on the line, placing A at origin: - A: 0 (origin) - AF = 73, so F: 73 - AC = 26, so C: 26 - CE = 31, so E: 26 + 31 = 57 - DF = 33, so D: 73 - 33 = 40 - BD = 22, so B: 40 - 22 = 18
[...detailed analysis omitted...]
2) Next, let's find coordinates of G using distances CG = 40 and DG = 30: - From C(26,0): $(x_G - 26)^2 + y_G^2 = 1600$ ...(1) - From D(40,0): $(x_G - 40)^2 + y_G^2 = 900$ ...(2)

Let's verify this using Python:

Therefore, the coordinates of the triangle vertices are: - B(18,0) - G(58,24) - E(57,0)

3) Calculate the area using the shoelace formula: Area $= \frac{1}{2}|x_1(y_2 - y_3) + x_2(y_3 - y_1) + x_3(y_1 - y_2)|$

Where: - $(x_1, y_1) = (18, 0)$ - $(x_2, y_2) = (58, 24)$ - $(x_3, y_3) = (57, 0)$

Substituting:

$$\text{Area} = \frac{1}{2}|18(24 - 0) + 58(0 - 0) + 57(0 - 24)|$$

$$= \frac{1}{2}|432 + 0 - 1368|$$

$$= \frac{1}{2}|-936|$$

$$= 468$$

Therefore, the area of triangle BGE is $\boxed{468}$.

## F.3 Test case of Hint-Engineering-RFT-32B

**A Case form AIME24**

***Problem:***
Let $\mathcal{B}$ be the set of rectangular boxes with surface area $54$ and volume $23$. Let $r$ be the radius of the smallest sphere that can contain each of the rectangular boxes that are elements of $\mathcal{B}$. The value of $r^2$ can be written as $\frac{p}{q}$, where $p$ and $q$ are relatively prime positive integers. Find $p + q$.

***Solution:***
The radius of the smallest sphere that can contain any rectangular box must be half of the maximum space diagonal. For a box with dimensions $a$, $b$, $c$, we have:

1. Space diagonal $= \sqrt{a^2 + b^2 + c^2}$ 2. Surface area: $2(ab + bc + ac) = 54$ 3. Volume: $abc = 23$

Our goal is to maximize $a^2 + b^2 + c^2$ subject to these constraints, then $r^2 = \frac{\max(a^2+b^2+c^2)}{4}$.

Let's try assuming $a = b$ to simplify the problem: - Surface area becomes: $2(a^2 + 2ac) = 54 \rightarrow a^2 + 2ac = 27$ - Volume becomes: $a^2c = 23 \rightarrow c = \frac{23}{a^2}$

[...detailed analysis omitted...]

Let's solve the resulting cubic equation:

**Python Code 1**

```python
from sympy import symbols, solve, Eq

a = symbols('a')
equation = Eq(a**3 - 27*a + 46, 0)
solutions = solve(equation, a)
print("Solutions:")
for sol in solutions:
    print(sol.simplify())
```

**Output 1**

```
Solutions:
2
-1 + 2*sqrt(6)
-2*sqrt(6) - 1
```

The only valid solution is $a = 2$, giving us: - $a = b = 2$ - $c = \frac{23}{4} = 5.75$

Let's verify this is indeed the maximum using numerical optimization:

**Python Code 2**

```python
from scipy.optimize import minimize

def objective(x):
    a, b, c = x
    return -(a**2 + b**2 + c**2)

def constraint1(x):
    a, b, c = x
    return 2*(a*b + b*c + a*c) - 54

def constraint2(x):
    a, b, c = x
    return a*b*c - 23

x0 = [2, 2, 5.75]
cons = [{'type': 'eq', 'fun': constraint1},
        {'type': 'eq', 'fun': constraint2}]
bounds = [(0.0001, None), (0.0001, None), (0.0001, None)]

result = minimize(objective, x0, method='SLSQP',
                  bounds=bounds, constraints=cons)

print("Optimization result:", result.success)
print("Optimal a, b, c:", result.x)
print("Maximum a² + b² + c²:", -result.fun)
```

Therefore: 1. Maximum space diagonal squared $= \frac{657}{16}$ 2. $r^2 = \frac{657}{16}/4 = \frac{657}{64}$ 3. $657 = 3 \times 3 \times 73$ and $64 = 2^6$ are coprime 4. Thus, $p = 657$ and $q = 64$

The answer is $p + q = 657 + 64 = \boxed{721}$.

---

### A Case from AIME25

***Problem****:*

Six points $A, B, C, D, E$ and $F$ lie in a straight line in that order. Suppose that $G$ is a point not on the line and that $AC = 26$, $BD = 22$, $CE = 31$, $DF = 33$, $AF = 73$, $CG = 40$, and $DG = 30$. Find the area of $\triangle BGE$.

***Solution****:*

First, let's assign coordinates to all points on the line, placing A at origin:

Given distances: - $AC = 26$ - $BD = 22$ - $CE = 31$ - $DF = 33$ - $AF = 73$

This allows us to determine coordinates: - A: $(0, 0)$ - B: $(18, 0)$ - C: $(26, 0)$ - D: $(40, 0)$ - E: $(57, 0)$ - F: $(73, 0)$

[...detailed analysis omitted...]

To find G's coordinates, we use distances $CG = 40$ and $DG = 30$:

1) From C $(26, 0)$: $(x_G - 26)^2 + y_G^2 = 1600$ ...(1) 2) From D $(40, 0)$: $(x_G - 40)^2 + y_G^2 = 900$ ...(2)

Let's solve this system using Python:

**Python Code 1**

```
from sympy import symbols, Eq, solve

x_G, y_G = symbols('x_G y_G')

# Equations based on CG = 40 and DG = 30
eq1 = Eq((x_G - 26)**2 + y_G**2, 1600)
eq2 = Eq((x_G - 40)**2 + y_G**2, 900)

# Solve for x_G and y_G
subtracted_eq = Eq((x_G - 26)**2 - (x_G - 40)**2, 700)
x_G_val = solve(subtracted_eq, x_G)[0]
y_G_val = solve(eq2.subs(x_G, x_G_val), y_G)[0]

print(f"G coordinates: ({x_G_val}, {y_G_val})")
```

**Output 1**

```
G coordinates: (58, 24)
```

Therefore, we have all coordinates: - B: $(18, 0)$ - G: $(58, 24)$ - E: $(57, 0)$

Let's calculate the area using the shoelace formula:

```
Python Code 2

# Calculate area using shoelace formula
B_coord = (18, 0)
G_coord = (58, 24)
E_coord = (57, 0)

area = 0.5 * abs(
    B_coord[0] * (G_coord[1] - E_coord[1]) +
    G_coord[0] * (E_coord[1] - B_coord[1]) +
    E_coord[0] * (B_coord[1] - G_coord[1])
)

print(f"Area of triangle BGE: {area}")
```

```
Output 2

Area of triangle BGE: 468.0
```

Verification using Base × Height method: - Base (BE) = $57 - 18 = 39$ - Height = $24$ - Area $= \frac{39 \times 24}{2} = 468$

Therefore, the area of triangle BGE is $\boxed{468}$.

## G  Strong-to-Weak Distillation Data

For the strong-to-weak distillation process, we constructed a comprehensive mathematical problem dataset of approximately 10,000 problems from three primary sources. The dataset comprises 800 problems from AIME (American Invitational Mathematics Examination) competitions prior to 2024, 2,280 problems from the MATH [67] dataset's training set (Level 5), and 7,000 problems sampled from the Numina-math [68] collection. For the Numina-math sampling, we implemented several filtering criteria: problems containing figures were excluded, as were proof-based problems, multiple-choice questions, and problems where answers appeared within the problem statements. After applying these filtering criteria, we randomly sampled 7,000 problems to ensure a clean and standardized dataset suitable for our distillation training process.

## H  Baselines

We compare our approaches against a range of state-of-the-art mathematical reasoning models across different parameter scales:

**SOTA Models.** We benchmark against the most advanced large reasoning models currently available: OpenAI's o1 [5], QwQ-32B [69], and DeepSeek-R1 [7]. These models represent the frontier of mathematical reasoning capabilities and serve as an upper bound for performance comparison.

**Frontier Models (32B).** For the 32B parameter scale, we use DeepSeek-R1-32B [7] as our foundation model since it serves as the starting point for many open-source models in this size range. We also compare against contemporary tool-integrated reasoning (TIR) models, including START-32B [21], STILL-3-TOOL-32B [17], and ReTool-R1-32B [60], the latter being a concurrent work focusing on reinforcement learning for tool use.

**Lightweight Models (1.5B).** In the lightweight category, we use DeepSeek-R1-1.5B [7] as our base model for the same reason as its 32B counterpart. We benchmark against DeepScaleR-1.5B-Preview [65], the current state-of-the-art reinforcement learning model at this scale, and ToRL-1.5B [58], a concurrent work on tool-integrated reasoning with reinforcement learning.

Our selection of baselines spans different model sizes, training methodologies (SFT, RL, and RFT), and approaches to mathematical reasoning (with and without explicit tool use). This comprehensive comparison allows us to evaluate the effectiveness of our Prompt-Hint and Hint-Engineering approaches relative to both established benchmarks and recent innovations in the field.

# I  Out-of-Distribution Generalization on Chemistry Problems

To rigorously evaluate the generalization capabilities of our CoRT framework, we conducted a challenging out-of-distribution (OOD) experiment on a domain far removed from our mathematical training data: **chemistry**. We utilized a subset of problems from the GPQA benchmark [70], which requires deep domain knowledge and specialized computational tools.

This experiment serves as a stringent test for several reasons:

1. **Domain Shift**: The problems involve reasoning about molecular structures and properties, a domain conceptually distinct from the algebra, geometry, and number theory problems used in training.
2. **Tool Shift**: Standard mathematical libraries like `NumPy` or `SymPy` are inadequate for these tasks. The gold-standard tool for cheminformatics is the **RDKit** library.
3. **Zero-Shot Tool Discovery**: Crucially, the `RDKit` library was **never included in any of our training data or prompts**. The model's ability to successfully solve these problems hinges on its capacity to autonomously identify the need for a new tool, discover the correct library (`RDKit`), and learn to use its functions on the fly.

We compared our `Hint-Engineering-RFT-32B` model against the baseline `DeepSeek-R1-32B` on a set of chemistry problems from GPQA. The evaluation focused on final answer accuracy, token efficiency, and, most importantly, the model's ability to spontaneously utilize the unseen `RDKit` library.

## I.1  Results and Analysis

The results, summarized in Table 3, demonstrate that our CoRT framework imparts a generalizable reasoning ability that successfully transfers across domains and tools.

Table 3: Performance on the OOD Chemistry (GPQA) Benchmark. Our model not only improves accuracy and efficiency but also spontaneously discovers and utilizes the unseen `RDKit` library, demonstrating true generalization.

| Model | Accuracy | Avg. Tokens | RDKit Usage Rate |
|---|---|---|---|
| DeepSeek-R1-32B (Baseline) | 40.6% | 5,947 | 0% |
| Hint-Engineering-RFT-32B | **47.5%** | **4,220** | **81.3%** |
| **Improvement** | **+6.9 pts** | **-29.0%** | **+81.3 pts** |

Our key findings from this experiment are:

- **Robust Cross-Domain Performance**: Our model achieved a **6.9 absolute point accuracy gain** over the baseline in the unseen chemistry domain. This provides powerful evidence that CoRT teaches a fundamental problem-solving methodology rather than domain-specific memorization.
- **Spontaneous Tool Discovery and Utilization**: The most striking result is that our model spontaneously discovered and correctly utilized the `RDKit` library in **81.3%** of cases, whereas the baseline model completely failed to do so. This showcases a sophisticated, emergent ability to adapt to new problem contexts, a feat impossible for systems reliant on fixed, pre-defined API sets.
- **Sustained Efficiency Gains**: Consistent with our in-domain results, the model also demonstrated significant efficiency improvements, using **29.0% fewer tokens** on average. This indicates that the learned behavior of prioritizing computation via code is a generalizable strategy.

In conclusion, this OOD experiment provides strong evidence that the reasoning patterns instilled by our Hint-Engineering and post-training pipeline are not brittle but foster a flexible and adaptive problem-solving capability, which is a critical step towards more general-purpose AI reasoning systems.

## J  Broader Impacts

Our work on enhancing code-integrated reasoning capabilities in LRMs offers several potential benefits to society. The improved efficiency and accuracy in mathematical reasoning could enhance educational applications and academic research, while reduced token usage contributes to environmental sustainability through decreased computational resource consumption. The integration of precise computational tools with natural language reasoning may also improve the reliability of AI systems in applications requiring accurate calculations.

However, as with any advancement in AI capabilities, potential risks should be considered. While our models are trained on public mathematical datasets and intended for educational and research purposes, the enhanced reasoning capabilities could potentially be misused if not properly secured. We recommend implementing appropriate access controls and maintaining transparency about the system's limitations and intended use cases.

We encourage future work to continue exploring responsible deployment strategies that maximize beneficial impacts while minimizing potential risks.

## K  Limitation

Our work presents a novel framework for enhancing code-integrated reasoning in LRMs through hint-engineering and efficient post-training strategies. While we demonstrate significant improvements in both performance and efficiency, particularly in mathematical reasoning tasks, several directions remain for future exploration. Due to our use of DeepSeek-R1-Distill-Qwen models (32B and 1.5B), the computational requirements for training and inference may affect broader accessibility. While we utilize publicly available datasets and models, ensuring our work maintains transparency and reproducibility, the rapid evolution of LRM capabilities suggests opportunities for continued enhancement as new architectures emerge. Furthermore, while our current work emphasizes the integration of code interpreters within long-form reasoning, there remain opportunities to investigate additional synergies between external tools and language models' inherent reasoning capabilities. As the field of large reasoning models continues to advance rapidly, we anticipate further developments in optimizing these interactions.

