# OpenReview forum: "Teaching Language Models to Reason with Tools"
_NeurIPS.cc/2025/Conference — NeurIPS 2025 poster_

### Official Review · Reviewer_jUL7 · 2025-06-17

**Clarity:** 2
**Significance:** 1
**Originality:** 1
**Rating:** 4
**Confidence:** 4

**Summary:**

This paper introduces prompting techniques to make the best use of code interpreter tools during the reasoning phase of modern LLMs.

The authors identify two inefficiencies in reasoning traces of modern LLMs: (1) they tend to prefer natural language computation and using Python as a verification tool only, (2) they verify again in natural language the code execution results. To mitigate these two limitations, the authors try to identify these moments in the reasoning sequence and inject “hints” to redirect the LLM towards (1) executing Python computations or (2) not verifying the accuracy of the Python calculation, respectively.

They generate multiple samples on the STILL3 dataset and filter out samples with incorrect answers or samples that don’t adhere to the above hints to rejection finetune a deepseek-r1-distill-qwen-32B with this data (resulting in the hint-engineering-rft-32B model).
They also use another prompt at the beginning of each reasoning sequence to tell the model to use multiple Python code calls and finetune a deepseek-r1-distill-qwen-32B with similar data (resulting in the prompt-hint-sft-32B model).

To conduct RL experiments, the authors distill their finetuned 32B models into 1.5B models by sampling from them on a diverse set of 10,000 examples.

Finally, the result section shows that the specially prompted models achieve comparable performance to other baselines of similar sizes but generates less tokens, hence is more efficient.

**Questions:**

### Questions

Q1: How do you manually inject hints to mitigate the two inefficiencies mentioned on page 4? Do you mean that you have a heuristic computation that checks if either of the two situations arises?

> A critical challenge in our approach was identifying suitable positions for hint insertion. [...]. Consequently, we opted for manual hint placement and insertion.

This sentence should be much more detailed, especially since this is a big part of the contribution of this paper.

Q2: At the end of section 2.2.2, you mention the “Hint-Engineering-SFT-32B model” and the D_{hint-engineering-sft} dataset, but these are not described before. What are these?

Q3: Why did you distill your fine-tuned 32B model into 1.5B models instead of finetuning directly the 1.5B models with the same data you used to train the 32B models?

### Typos
- Line 60: [...] to encourage writes the codes correctly.
- Line 75: [...] as illustrated in Figure 2.1 – there is only figure 2.
- Line 145-150: duplicate wording in line 145 & 149. Not coherent for line 145 but ok for line 149.

**Ethical Concerns:**

["NO or VERY MINOR ethics concerns only"]

**Final Justification:**

Based on the provided explanations and additional studies, I will raise my score from 2 to 4, however, I request that the authors must include (1) clarification about the data synthesis framework (2) clarification about the usage of the RL safeguard hint, and (3) clarification about the manual annotation workflow in the main text of the paper, along with a literature review.

**Limitations:**

No limitation discussion.
One limitaion of this work is the focus on math problems and only the python interpreter tool. Future work can explore more diverse domains.
A brief paragraph on the societal impacts of having better llms at math and overall tool use could be included.

**Paper Formatting Concerns:**

ok

**Quality:**

2

**Strengths And Weaknesses:**

## Strengths

This paper raises an important point as modern LLMs having access to a panoply of external tools should be able to use them efficiently. Experiments are well made and empirical analysis show how efficient the proposed prompt can be.

## weaknesses

W1: The paper omits several important aspects in the main text and instead sends the reader in the appendix. While it is acceptable to save space by puting **additional results** in appendices, the main experimental setup (baselines, training objectives, data splits, metrics, etc…) should be part of the main text. Similarly for the literature review, while a longer version can be in the appendix, a short version should still be in the main text.
In its current form, the paper is missing important details making it hard to understand (see questions in the Questions section).

W2: The data synthesis framework is presented as a core contribution of this paper; however it is not clear what it is? The authors generated 30 samples manually and prompted LLMs to generate more samples on STILL3. Is the “framework” how the 30 example were generated? If so, then while quality is indeed more important than quantity, a data synthesis framework should still be scalable to more than 30 examples. Is the “framework” related to how the LLMs are prompted? If so, calling it a prompt rather than a framework would be more accurate.

W3: In the RL experiments, injecting a “hint” to say to the model to **not** use Python anymore contradicts previous hints in the prompt. In addition, this adhoc intervention will directly impact the conclusion you will observe to the research question of _“How the models’ interaction patterns with the Code Interpreter evolve during the RL process?”_

---

> ### Author Rebuttal · Authors · 2025-07-31
>
> We sincerely thank the reivewer for their thorough review and constructive feedback. We are particularly grateful that the reviewer recognizes the importance of our research direction—**enabling LLMs to use external tools efficiently**—and acknowledges that our **"experiments are well made and empirical analysis show how efficient the proposed prompt can be."** This encouragement is invaluable.
>
> We are grateful for the opportunity to address the remaining concerns:
>
> ---
>
> **W1: Details in Main Text**
>
> Thank you for this crucial feedback. We agree completely. Due to strict page limits, we initially prioritized presenting our key findings in the main text while moving detailed setups to the appendix. In the revision, we will integrate these essential details and a concise literature review into the main text, relocating less critical material to the appendix.
>
> **W2: The Data Synthesis Framework**
>
> Thank you for seeking clarification. We understand the confusion and appreciate the opportunity to elaborate on this core contribution. Our "data synthesis framework" is a **systematic, scalable pipeline** designed to teach models efficient tool-integrated reasoning. Its necessity and effectiveness are rooted in a two-stage process:
>
> 1.  **Stage 1: Seed with High-Quality Data (Hint-Engineering & SFT).** We start by manually creating 30 high-quality "seed" examples. This small dataset is **essential** because, as we will demonstrate below, prompting alone **cannot** teach the model the desired efficient reasoning patterns. These 30 examples provide focused demonstrations of how to proactively use code for computation, not just verification. **In this stage, we fine-tune the base `DeepSeek-R1-32B` model on these 30 seed examples to produce our `Hint-Engineering-SFT-32B` model.**
>
> 2.  **Stage 2: Scale with Automated Synthesis (RFT).** After SFT on the seed data "unlocks" the correct behavior, we use the now-improved model to generate thousands of candidate solutions. These are then automatically filtered based on correctness and reasoning quality to create a larger, high-quality dataset for Rejection Fine-Tuning (RFT). This creates a self-improving cycle that scales our data from 30 to 830 examples (with potential for further expansion given additional computational resources). **In this stage, the `Hint-Engineering-SFT-32B` model serves as the input, and after RFT on the scaled data, we obtain our final `Hint-Engineering-RFT-32B` model.**
>
> To prove our seed data is necessary, we ran a zero-shot experiment on the base model using a strong TIR prompt without fine-tuning. The results below show our framework teaches a superior reasoning paradigm that prompting alone cannot achieve.
>
> | Model                     | Dataset | Avg. Accuracy | Avg. Tokens | Avg. Code Blocks |
> | :------------------------------------- | :---------- | :---------------: | :-----------------------: | :-------------------------------: |
> | TIR Prompt-hint on DeepSeek-R1-32B    | AIME24      |       67.9%       |           9,382           |             0.89             |
> |                                        | AIME25      |       50.4%       |          11,902           |            0.68              |
> | Hint-Engineering-RFT-32B | AIME24      |    **76.3%**      |   **7,260** `(-22.6%)`   |             **2.10**              |
> |                                        | AIME25      |    **65.2%**      |   **8,532** `(-28.3%)`   |             **2.64**              |
>
> **Key Findings:**
> *   **Superior Performance & Efficiency:** Our model achieves significantly higher accuracy (**+14.8 pts** on AIME25) with substantially fewer tokens (**-28.3%**).
> *   **Fundamental Behavioral Shift:** Our method increases code usage by **2-4x**, shifting the model from an inefficient "verify-with-code" pattern to an efficient "compute-with-code" strategy. This demonstrates our data is both necessary and effective.
>
> **W3: RL Safeguard Mechanism**
>
> We appreciate this sharp observation. The hint to stop using Python is **not a contradictory instruction but a critical computational safeguard**, triggered only when a model exceeds the predefined 15-tool-call limit. This is a standard practice to prevent infinite loops and manage resources during large-scale RL, also utilized in works like the AIMO-2 winning solution by NVIDIA [1].
>
> Crucially, this safeguard has a **negligible impact on our conclusions** because it affects only a tiny fraction of failing cases. Our analysis of the RL training logs provides clear evidence: the limit was reached in only **0.2%** of all trajectories. Furthermore, among these rare instances, **85.6%** involved models already trapped in non-productive error loops.
>
> This data shows the intervention is a rare mechanism for exceptional failure modes and does not influence learned behavior in the vast majority of trajectories.
>
> [1] "AIMO-2 Winning Solution: Building State-of-the-Art Mathematical Reasoning Models with OpenMathReasoning dataset"
>
> ---
>
> **Q1: Manual Hint Injection**
>
> Our process relies on expert judgment guided by a replicable procedure—not on heuristics. We adopted this manual approach because automated methods failed to produce satisfactory results. Given the complexity of our task, we are unaware of any purely automated methods that can succeed without human insight.
>
> **Why Manual Curation was Necessary:** We initially used powerful LLMs as judges to automate hint insertion. However, they consistently failed on subtle but critical distinctions. On a test set of 100 AIME problems, we identified two primary failure modes:
>
> | Failure Mode                                    | Description                                                                                                                              | Occurrence |
> | :-------------------------------------------------- | :------------------------------------------------------------------------------------------------------------------------------------------- | :------------: |
> | 1. Confusing Planning with Inefficient Calculation | The judge struggled to differentiate between necessary high-level planning and low-level calculations that should be offloaded to the CI.          |    74/100      |
> | 2. Confusing Logical Verification with Numerical Distrust | The judge couldn't distinguish between a valid check of code logic versus unnecessarily re-calculating a trusted CI output. |    19/100      |
>
> These frequent and critical errors led us to conclude that a small, high-precision, manually-annotated seed dataset was the most effective way to bootstrap the learning process.
>
> **The Manual Workflow:**
> Our manual editing follows a clear, three-step procedure:
>
> 1.  **Generate Baseline Trace:** An annotator prompts the base model to generate a complete reasoning trace for a problem.
> 2.  **Identify Inefficiencies:** The annotator inspects the trace, looking for two specific, easily identifiable patterns of redundancy:
> 	*   **Delayed Computation ("Thinking too hard"):** The model first guesses an answer in text (e.g., solving `x^2 - 17x + 60 = 0` by sight) before using code (`sympy.solve(...)`) for a precise result. The initial guess is inefficient and error-prone.
> 	*   **Code Distrust ("Not trusting the calculator"):** After getting a precise code output (e.g., `>>> [5, 12]`), the model re-verifies it in text (e.g., plugging roots back into the equation). This is redundant.
> 3.  **Surgical Correction:** Based on the identified inefficiency, the annotator performs a targeted edit. For "thinking too hard", they remove the text-based calculation and insert a "use code" hint earlier. For "not trusting the calculator", they remove the redundant verification and insert a "trust Python" hint.
>
> Note that prior works like InstructGPT also designed detailed annotation instructions. In the revision, we will clarify our own instructions for reference.
>
> **Q2: Model & Dataset Naming**
>
> We apologize for the lack of clarity.
> *   **D_{hint-engineering-sft}:** This is the dataset of **30 high-quality reasoning traces** created through the manual "Hint-Engineering" process described in our response to Q1. It is first mentioned in Line 132.
> *   **Hint-Engineering-SFT-32B:** This is the model resulting from **Supervised Fine-Tuning (SFT)** the base DeepSeek-R1-32B model on the `D_{hint-engineering-sft}` dataset. It serves as the starting point for our Rejection Fine-Tuning stage and is mentioned in Lines 128-129.
>
> **Q3: Distillation vs. Direct Fine-tuning**
>
> This is an excellent question. We chose distillation because direct fine-tuning the 1.5B model with our small dataset was **ineffective**, as it lacks the capacity to benefit from such subtle guidance. In contrast, the 32B model learns effectively and can then generate a larger dataset to teach the smaller model via distillation.
>
> The table below confirms this. Direct fine-tuning shows negligible improvement over the baseline, while our distillation approach provides meaningful gains.
>
> | Model (1.5B)         | Training Method    | Avg. Accuracy |
> | :----------------------- | :--------------------- | :---------------: |
> | DeepSeek-R1-1.5B         | Base Model (None)      |       48.1%       |
> | SFT on DeepSeek-R1-1.5B | Direct Fine-Tuning     |       43.2%       |
> | Hint-Eng-SFT-1.5B (Ours) | Distillation from 32B |     **51.2%**     |
>
> This approach effectively transfers capabilities and enables large-scale RL on smaller, computationally manageable models.
>
> ---
>
> We sincerely thank you for your insightful review. We gratefully hope that you could re-evaluate our paper based on the clarifications provided above. If our responses have satisfactorily addressed your concerns, we would greatly appreciate it if you could consider updating the review score accordingly.

---

> > ### Author Response · Authors · 2025-08-05
> >
> > Dear Reviewer jUL7,
> >
> > Thank you again for your thorough review and for raising several important questions for clarification. The discussion period has been a valuable opportunity for us to provide additional details and new results to further strengthen the paper for all readers.
> >
> > In that spirit, we hope our detailed rebuttal has fully addressed your specific points. We aimed to provide comprehensive clarifications by:
> > 1.  **Providing a step-by-step walkthrough of our two-stage "Data Synthesis Framework" (W2),** illustrating how we scale from the initial high-quality seeds to a large-scale dataset.
> > 2.  **Detailing the precise, replicable manual workflow for "Hint-Engineering" (Q1),** making our core data curation strategy transparent and easy to follow.
> > 3.  **Presenting a new ablation study (Q3)** that provides a clear, data-driven justification for our distillation strategy over direct fine-tuning.
> > 4.  Committing to integrating these crucial details into the main text in the revised version to enhance readability and accessibility.
> >
> > We believe these elaborations make the novelty, rigor, and effectiveness of our framework much more apparent. We would be very grateful if you could consider our comprehensive response before the discussion period closes on Aug 6.
> >
> > Please let us know if any questions remain.
> >
> > Best regards,
> >
> > The Authors

---

> ### Comment · Reviewer_jUL7 · 2025-08-05
> **thanks for the clarifications**
>
> Dear authors,
>
> Thank you very much for addressing most of my concerns and questions.
>
> 1. About the data synthesis framework: your explanation is very clear and this two step bullet point should be in the paper as it makes the contribution more clear. Thanks for that.
>
> 2. About the RL safeguard mechanism: only 0.2% of cases, then it is indeed negligible. it was not clear in the text that this hint was only used when the model exceeded the 15-tool-call limit. Such clarification should be in the main text.
>
> 3. Thank you for the very detailed explanation of the manual annotation workflow. This is again crucial information that should be in the paper. Having it as a list of bullet points, like here makes it very clear.
>
> Based on the provided explanations and additional studies, I will raise my score from 2 to 4, however, I request that the three points above must be in the main text of the paper, along with a literature review.

---

> > ### Author Response · Authors · 2025-08-05
> > **Thanks for the Response**
> >
> > Dear Reviewer jUL7,
> >
> > Thank you very much for your thoughtful feedback and for taking the time to carefully consider our rebuttal. We sincerely appreciate your positive re-evaluation of our work and are grateful that you found our explanations clear and convincing.
> >
> > We fully agree with your suggestions and will ensure that the following key points are explicitly included in the main text of the revised manuscript:
> > - A clear two-stage description of our **data synthesis framework** (seed data creation + automated scaling via RFT).
> > - A detailed explanation of the **RL safeguard mechanism**, specifying that the "stop using Python" hint is only triggered when the 15-tool-call limit is exceeded, and that this occurs in only 0.2% of trajectories.
> > - The **manual annotation workflow**, presented as a clear, step-by-step procedure (e.g., using bullet points) to enhance transparency and reproducibility.
> > - A concise **literature review** in the main text to better contextualize our work.
> >
> > These revisions will significantly improve the clarity and accessibility of our paper, and we are committed to incorporating them in the final version.
> >
> > Once again, thank you for your constructive comments and for recognizing the value of our contributions. We are happy to have had the opportunity to clarify our approach, and we appreciate your fair and open-minded reassessment.
> >
> > Best regards,
> > The Authors

---

### Official Review · Reviewer_mxrG · 2025-07-02

**Clarity:** 3
**Significance:** 3
**Originality:** 3
**Rating:** 4
**Confidence:** 4

**Summary:**

This paper proposes CoRT, a post-training framework for teaching LRMs to effectively integrate code interpreters for improving the model's ability in complex mathematical reasoning. The method introduces prompt-hint and hint-engineering strategies to synthesize high-quality code-integrated reasoning data, followed by supervised fine-tuning, rejection fine-tuning, and reinforcement learning. Experiments on multiple math benchmarks demonstrate the effectiveness of CoRT in improving both accuracy and efficiency in math reasoning.

**Questions:**

1. Will the proposed method cause the fine-tuned model to overuse the Code Interpreter, even when it is not needed?
2. What are the code triggering rates of the fine-tuned models?
3. In the abstract and introduction, the author mentioned that they manually created 30 high-quality examples to post-train models. But in line 134, it mentions that 830 examples are created to obtain the Hint-Engineering-RFT-32B model. Is it a typo here?
4. The proposed hint-engineering method encourages models to avoid reflecting on code execution results in order to reduce redundant verification and improve token efficiency. However, in cases where the model generates incorrect or semantically flawed code, would this suppression of reflection hinder the model's ability to detect and correct its own code errors?

**Ethical Concerns:**

["NO or VERY MINOR ethics concerns only"]

**Final Justification:**

I recognize the paper’s main contribution and the strength of its comprehensive experiments. Since the authors have adequately addressed the main concerns raised in the initial review and discussion, I maintain my positive score.

**Limitations:**

Yes.

**Quality:**

3

**Strengths And Weaknesses:**

Strengths:
1. The paper focuses on an important and timely topic, which is how to effectively integrate external tools like Code Interpreters into LRM to improve mathematical reasoning.
2. In general, the writing of the paper is clear, well-structured and easy to follow.
3. The proposed method is simple yet effective, using well-designed prompt strategies and fine-tuning techniques to enhance both accuracy and efficiency.

Weakness:
1. The paper does not provide comparisons with state-of-the-art tool-using models such as o3 and o4-mini. While these models are mentioned as motivation, it would strengthen the empirical analysis to include their performance as a reference.
2. While the paper emphasizes the goal of teaching LLMs when and how to use tools, the proposed method mainly relies on prompt-based heuristics and fine-tuning to induce tool usage. It lacks an explicit mechanism for learning the decision of whether to invoke the external tool, such as a gating function or planner module. As a result, the model may not develop a genuine understanding of tool invocation timing, limiting its flexibility and generalization to unseen scenarios.
3. There is a typo in line 267 "probelms"-->"problems".

---

> ### Author Rebuttal · Authors · 2025-07-31
>
> We sincerely thank the reviewer for their insightful and constructive feedback. We are particularly grateful that the reviewer recognizes the key strengths of our work, including the **importance and timeliness of the topic**, the **clarity and structure of our writing**, and the **simplicity yet effectiveness of our proposed CoRT framework**. Your comments have been invaluable in helping us refine our paper.
>
> We are grateful for the opportunity to address the remaining concerns and further clarify our contributions:
>
> ---
>
> **W1: Comparison with SOTA Models**
>
> We thank the reviewer for this excellent suggestion. We agree that including the performance of SOTA models like o3 and o4-mini provides a crucial point of reference.
>
> We will update Table 1 in our revised manuscript to include these comparisons. The updated section of the table will be as follows:
>
> | Model                                         | Tool-Use | AIME24 (Pass@1) | AIME25 (Pass@1) |
> | :-------------------------------------------- | :------: | :-------------: | :-------------: |
> | **Hint-Engineering-RFT-32B** |    ✓     |    76.7%    |    67.1%    |
> | o3                                            |    ✗     |      91.6%      |      88.9%      |
> | o3                                            |    ✓     |      95.2%      |      98.4%      |
> | o4-mini                                       |    ✗     |      93.4%      |      92.7%      |
> | o4-mini                                       |    ✓     |      98.7%      |      99.5%      |
>
> ---
>
> **W2 & Generalization: Lack of an Explicit Planner**
>
> We thank the reviewer for this excellent and thought-provoking question about our method's approach to learning tool invocation. The reviewer correctly notes that we do not use an explicit, separate module like a planner or a gating function. This is a deliberate design choice, and we believe our integrated, end-to-end approach fosters a more genuine and flexible understanding of tool use.
>
> 1.  **An Integrated, End-to-End Decision Process:** Instead of offloading the decision to a separate module, our framework teaches the LRM to make the tool-use decision as an intrinsic part of its own reasoning process. The model learns to generate the special `'''python` token at the appropriate time, based on the entire context of the problem and its reasoning chain. This is achieved through fine-tuning on high-quality exemplars (our "Hint-Engineering" data) where the timing of tool invocation is optimal. We argue this internalization is a more powerful form of learning than relying on a simpler, potentially brittle, external gate.
>
> 2.  **Flexibility and Contextual Nuance:** An explicit gating function often reduces the decision to a simplified binary choice. Our integrated approach is far more flexible. The decision to invoke the Code Interpreter is made by the full LRM, which can consider the nuance of the entire conversation history. It's not just a decision of *whether* to use a tool, but *which specific sequence of computations* is needed at that exact moment.
>
> 3.  **Empirical Evidence of Generalization:** Our framework's ability to develop a flexible and generalizable understanding is empirically supported by our results.
>     *   **Diverse Code Usage:** Our Code Behavior Analysis (Section 3.3, Figure 5) shows that the fine-tuned models learn to use a wide variety of Python functions for complex tasks like "Solving Equations," and "Symbolic Mathematics." This demonstrates a sophisticated, context-dependent application of the tool.
>     *   **Out-of-Distribution (OOD) Generalization:** To further address transferability, we conducted an OOD evaluation on **Chemistry problems from the GPQA benchmark**. This domain is far removed from our mathematical training data. Crucially, the standard tool for this domain, the `RDKit` library, **was never included in our training data**. The model's ability to spontaneously discover and effectively use this domain-specific library serves as a rigorous test of its generalization.
>
> | Model | Accuracy | Avg. Tokens | RDKit Usage Rate |
> | :--- | :---: | :---: | :---: |
> | DeepSeek-R1-32B (Base) | 40.6% | 5,947 | 0% |
> | Hint-Engineering-RFT-32B | 47.5% | 4,220 | 81.3% |
> | Improvement | **+6.9 pts** | **-29.0%** | **+81.3 pts** |
>
> These results strongly suggest that the model has generalized the decision-making process of *when* and *how* to invoke tools, rather than simply memorizing triggers.
>
> ---
>
> **Q1: Will the method cause tool overuse?**
>
> Thank you for this crucial question. Our analysis on the MATH500 dataset, stratified by difficulty, shows that our framework does not lead to indiscriminate overuse. Instead, it fosters an **adaptive and intelligent tool-use policy**, where code usage is modulated based on problem difficulty.
>
> The table below for our `Hint-Engineering-1.5B` model demonstrates this clearly:
>
> | MATH500 Level (Difficulty) | Avg. Code Blocks (after SFT) | Avg. Code Blocks (after RL) |
> | :--- | :---: | :---: |
> | **Level 1 (Easiest)** | 2.05 | 1.63 |
> | **Level 2** | 1.94 | 1.89 |
> | **Level 3** | 2.63 | 2.01 |
> | **Level 4** | 2.70 | 2.25 |
> | **Level 5 (Hardest)** | 2.71 | 2.58 |
> | **Trend** | **Usage increases with difficulty** | **Usage increases with difficulty** |
>
> As shown, the model consistently uses **fewer code blocks for simpler problems and more for complex ones**. This intelligent, demand-driven usage pattern is the antithesis of "overuse" and a hallmark of an efficient problem-solver.
>
> ---
>
> **Q2: What are the code triggering rates?**
>
> Our models achieve high and stable code triggering rates after training, demonstrating they have robustly learned to invoke the Code Interpreter.
>
> The code usage rate after Supervised Fine-Tuning (SFT) is already high, and it is further reinforced and stabilized during Reinforcement Learning (RL). The table below details the code usage rates on the AIME24 dataset at different stages of RL training for our 1.5B models.
>
> | Training Stage | Prompt-Hint-1.5B | Hint-Engineering-1.5B |
> | :--- | :---: | :---: |
> | After SFT (Step 0) | 76.7% | 86.9% |
> | RL Step 20 | 86.9% | 95.0% |
> | RL Step 40 | 94.4% | 97.7% |
> | RL Step 60 | 94.6% | 99.0% |
> | RL Step 80 | 97.5% | 99.0% |
> | RL Step 100 | 96.5% | 99.6% |
> | RL Step 120 | 97.1% | 97.9% |
>
> As the table shows, the **Hint-Engineering** approach achieves a very high triggering rate of **86.9%** immediately after SFT, which then quickly climbs to and stabilizes **above 95%** during RL. This indicates that the model has effectively learned to utilize the tool consistently. A full visualization of this trend, along with other code behavior metrics, is available in **Appendix D.1, Figure 8**.
>
> ---
>
> **Q3: Clarification on 30 vs. 830 examples.**
>
> We thank the reviewer for this question and for the careful reading. This is not a typo; the two numbers refer to two distinct datasets used in our sequential, multi-stage training pipeline designed to scale up high-quality data.
>
> Our process involves two main fine-tuning stages, where the output model of one stage becomes the input for the next:
>
> 1.  **Stage 1: Bootstrapping with SFT.** We start with the base `DeepSeek-R1-32B` model. We perform Supervised Fine-Tuning (SFT) on it using a small, manually created set of **30 high-quality samples** (`D_Hint-engineering-SFT`). The goal is to "bootstrap" the model, teaching it the fundamental principles of our efficient Hint-Engineering reasoning pattern. The output of this stage is the `Hint-Engineering-SFT-32B` model.
>
> 2.  **Stage 2: Scaling with RFT.** We then use the improved `Hint-Engineering-SFT-32B` model from Stage 1 to generate solutions for a larger set of 820 problems. After automatically filtering these generated solutions for correctness and quality, we obtain approximately 800 new high-quality trajectories. We combine these with our original 30 manual samples to create a larger, scaled-up dataset of **830 examples** (`D_Hint-engineering-RFT`). This dataset is then used for Rejection Fine-Tuning (RFT) on the `Hint-Engineering-SFT-32B` model, producing our final, most capable model: `Hint-Engineering-RFT-32B`.
>
> The full pipeline is summarized in the table below:
>
> | Fine-Tuning Stage | Input Model | Training Data Used | Data Size | Data Origin | Output Model |
> | :--- | :--- | :--- | :---: | :--- | :--- |
> | **SFT** | DeepSeek-R1-32B | `D_Hint-eng-SFT` | 30 | Manually created & annotated | Hint-Eng-SFT-32B |
> | **RFT** | Hint-Eng-SFT-32B | `D_Hint-eng-RFT` | 830 | 30 manual + 800 generated & filtered | **Hint-Eng-RFT-32B** |
>
> ---
>
> **Q4: Does suppressing reflection hinder self-correction?**
>
> We thank the reviewer for this crucial question. Our method is carefully designed to avoid this exact problem by making a critical distinction between two types of reflection:
> *   **Unnecessary Numerical Verification:** Doubting the computational accuracy of the Code Interpreter (e.g., checking if `2+2` is `4`). Our hints are designed to suppress this inefficient behavior.
> *   **Essential Logical Verification:** Checking the semantic correctness of the model's *own generated code* to ensure it aligns with the problem-solving plan. This is a vital self-correction skill that we preserve and encourage.
>
> As we explicitly state in the paper on lines 122-124, *"while we discourage the verification of Python’s numerical calculations, we maintain the model’s behaviour to verify the logical correctness of the code structure"*. Therefore, our approach improves efficiency without hindering the model's ability to detect and correct its own logical code errors.
>
> ---
>
> **Typo Correction:**
>
> Thank you for catching the typo in line 267. We will correct this to "problems" in the revised manuscript.
>
> ---
>
> Thank you again for your valuable and encouraging feedback. We believe that incorporating these changes will significantly strengthen our paper. Please let us know if there is anything else we can do to help you better understand and recommend our work.

---

> > ### Comment · Reviewer_mxrG · 2025-08-04
> > **Thank you for your rebuttal**
> >
> > Thanks for the authors' detailed response. My main concerns are addressed and I will maintain the positive score for the paper.

---

> > > ### Author Response · Authors · 2025-08-05
> > >
> > > Thank you very much for your thoughtful review and for your quick and positive response to our rebuttal. We are delighted to hear that your main concerns have been addressed.
> > >
> > > We are especially grateful for your appreciation of the importance of our topic, the clarity of our writing, and the effectiveness of the CoRT framework. Your recognition of these key strengths means a great deal to us.
> > >
> > > Thank you again for your valuable time and support.

---

### Official Review · Reviewer_xQk3 · 2025-07-02

**Clarity:** 3
**Significance:** 2
**Originality:** 2
**Rating:** 4
**Confidence:** 3

**Summary:**

This paper introduces CoRT, a framework designed to teach LRMs how to effectively use a Code Interpreter (CI) for mathematical reasoning. The authors identify key inefficiencies in existing models, namely "delayed code computation" and "code result distrust." To address this, they propose "Hint-Engineering," a method of manually inserting strategic hints into reasoning traces to guide the model toward more efficient use of the CI. Based on a small set of 30 manually crafted examples, they fine-tune models of various sizes and demonstrate absolute accuracy improvements and significant reductions in token consumption (up to 50%) on several mathematical benchmarks.

**Questions:**

1. The abstract and main results table highlight specific absolute performance gains of 4% and 8% for your models. Given that the paper acknowledges these results are from single experimental runs due to computational costs, how can you ensure these improvements are statistically significant and not a result of random factors like initialization?



2. Your core contribution, "Hint-Engineering," relies on manually creating 30 high-quality samples and manually inserting hints, because you found that attempts to automate this process were "insufficiently precise". Can you elaborate on the specific failure modes of your automation attempts and explain how this framework could be scaled to new or larger problem domains without requiring prohibitive amounts of expert manual labor?

**Ethical Concerns:**

["NO or VERY MINOR ethics concerns only"]

**Final Justification:**

The authors have addressed my concerns in the rebuttal.

**Limitations:**

See the questions.

**Paper Formatting Concerns:**

seems no issues

**Quality:**

2

**Strengths And Weaknesses:**

## Minor Concerns
- Overstated Novelty: The paper frames CoRT as a new framework. However, the constituent parts—SFT, RFT, strong-to-weak distillation, and RL on curated data—are all established techniques. The primary novelty lies in the manual "Hint-Engineering" data curation strategy, whose limitations are discussed below.


- Token Efficiency Framing: The paper highlights a 30-50% reduction in token usage. While this is a valuable practical outcome, it is a direct and intended result of the hint design, which explicitly discourages textual reasoning in favor of code execution. This is more of a behavioral shift guided by data curation than a fundamental improvement in the model's innate reasoning efficiency

## Major Concerns

- The most significant flaw in this paper is the absence of any statistical validation for the empirical results. The authors explicitly state in the checklist (Question 7) that they did not report error bars or conduct multiple runs due to the high computational cost of LLM inference. While the resource constraints are understandable, a single experimental run is insufficient to conclude that the observed gains are not the result of random factors such as initialization seed or other sources of noise. This lack of rigor undermines the paper's central claims, such as the 4% and 8% absolute improvements cited in the abstract. For instance, the average accuracy difference between the main proposed model, Hint-Engineering-RFT-32B (81.3%), and the Prompt-Hint-SFT-32B model (81.8%) is small and could easily fall within a standard margin of error.

- Limited Scalability and Generalizability of "Hint-Engineering": The core methodological contribution, "Hint-Engineering," is described as a process of manually creating 30 high-quality samples and manually inserting hints. The authors concede that their attempts to automate hint placement were "insufficiently precise". This heavy reliance on manual, expert-driven intervention raises serious questions about the scalability and generalizability of the proposed framework. The contribution appears to be less of a novel, scalable training method and more of a highly specific, labor-intensive data curation effort. It is unclear how this approach would be applied to new domains or larger, more diverse datasets without a similar, prohibitive level of manual effort. The paper fails to provide a path toward automating this critical step, which is a significant limitation.

---

> ### Author Rebuttal · Authors · 2025-07-29
>
> We sincerely thank the reviewer for their detailed and insightful feedback. We are encouraged that the reviewer recognizes the value of our work, particularly the significant practical outcome of achieving a **30-50% reduction in token consumption** while improving accuracy, and acknowledges that our primary novelty lies in the **"Hint-Engineering" data curation strategy**. We appreciate the opportunity to clarify our contributions and address the concerns raised.
>
> ---
>
> **1. On the Novelty of the CoRT Framework**
>
> We acknowledge that the individual techniques—SFT, RFT, strong-to-weak distillation, and RL—are well-established. Our novelty lies not in inventing these components, but in **systematically combining them into a data-centric framework specifically engineered for the unique challenges of code-integrated reasoning.**
>
> *   **Addressing a Core Weakness:** Our work tackles a fundamental limitation of LRMs: their struggle with precise numerical calculations and complex, multi-step logic. By integrating a Code Interpreter (CI), we augment the model's internal reasoning with external computational power.
> *   **Unique Challenges of Tool-Integrated Reasoning:** This interactive, tool-augmented reasoning is fundamentally different from tasks that rely solely on a model's internal knowledge. It introduces new challenges, such as deciding *when* to use a tool, *how* to formulate the query, and *how* to interpret the feedback.
> *   **Principled Data-Centric Approach:** Through systematic data curation, we identified specific failure modes—such as "delayed code computation" and "code result distrust"—and developed targeted data patterns to address them. This principled approach to designing data for tool-integrated reasoning provides actionable insights for this emerging domain.
>
> Our contribution is demonstrating *how* to effectively orchestrate existing techniques to teach a model a new, more efficient reasoning paradigm, a critical step toward building more capable and practical AI systems.
>
> ---
>
> **2. On the Framing of Token Efficiency and the Necessity of Data**
>
> We agree with the reviewer that the token reduction is a direct and intended result of our hint design. However, we respectfully argue that this guided "behavioral shift" is a significant contribution. **Achieving this shift is non-trivial, as simply instructing a model to "use code more" is often ineffective.**
>
> To substantiate this, we conducted an experiment demonstrating that our small-scale, high-quality data is **essential** for teaching this superior reasoning paradigm. We compared our fine-tuned `Hint-Engineering-RFT-32B` model against the `DeepSeek-R1-32B` model using only zero-shot prompting (without any fine-tuning).
>
> | **Model / Method** | **Dataset** | **Avg. Accuracy** | **Avg. Tokens** | **Avg. Code Blocks per Response** |
> | :--- | :--- | :---: | :---: | :---: |
> | TIR Prompting on DeepSeek-R1-32B | AIME24 | 67.9% | 9,382 | 0.89 |
> | | AIME25 | 50.4% | 11,902 | 0.68 |
> | Hint-Eng-RFT-32B | AIME24 | **76.3%** | **7,260** `(-22.6%)` | **2.10** |
> | | AIME25 | **65.2%** | **8,532** `(-28.3%)` | **2.64** |
>
> **Key Findings:** Our method significantly outperforms the zero-shot baseline in accuracy (**up to +14.8%**) and token efficiency (**-23-28%**). The model learns a new reasoning strategy, increasing code usage by **2-4x** for proactive computation. This demonstrates that fine-tuning on our data instills a superior, more efficient behavior unattainable through prompting alone.
>
> ---
>
> **3. On Statistical Validation for Performance Gains (Major 1 & Q1)**
>
> We thank the reviewer for highlighting this critical point and apologize if our initial presentation was unclear. To directly address your question about ensuring the **4% and 8% absolute improvements** are statistically significant, we provide the following clarifications.
>
> *   **Multiple Runs Are Incorporated:** Our experimental setup already incorporates multiple runs for statistical validity, following best practices of DeepSeek-R1. As stated in the caption of Table 1 of the paper, results are indeed averages:
>     *   **AIME24, AIME25, and AMC23:** Averaged over **16 samples** per problem.
>     *   **MATH500 and Olympiad:** Averaged over **4 samples** per problem.
>
> *   **Statistical Significance Testing:** To further validate our claims, we conducted **pairwise Wilcoxon signed-rank tests** on a per-problem basis by pooling results across benchmarks. This provides a robust micro-average statistical analysis.
>
> | Model Scale | Model A (Ours) | Model B (Baseline) | Acc. Improv. | p-value (Acc) | Token Reduction | p-value (Tokens) |
> |:---:|:---|:---|:---:|:---:|:---:|:---:|
> | 32B | Hint-Eng-RFT | DeepSeek-R1 | +3.80% | 0.055 | 36.3% | < 0.001 |
> | 1.5B | Hint-Eng-RL | DeepSeek-R1 | +7.41% | 0.013 | 59.1% | < 0.001 |
>
> **This analysis confirms:** The accuracy gains are statistically significant for the 1.5B model (**+7.41%, p=0.013**) and show a strong positive trend for the 32B model (**+3.80%, p=0.055**). Token reduction is highly significant (**p < 0.001**) for both, providing robust empirical support for our claims.
>
> ---
>
> **4. On Scalability and Generalizability of "Hint-Engineering" (Major 2 & Q2)**
>
> We appreciate the reviewer's concern. To address your question about scalability, we clarify (1) why manual curation was initially necessary and (2) how our framework is designed to be scalable and generalizable.
>
> **Why Manual Curation? Elaboration on Automation Failures**
>
> Our initial attempts to fully automate hint insertion using powerful LLM judges failed due to a lack of precision. On a test set of 100 AIME problems, we identified two primary failure modes:
>
> | Failure Mode | Description | Example Failure | Occurrence |
> | :--- | :--- | :--- | :---: |
> | **1. Distinguishing Planning from Inefficient Calculation** | The judge struggled to differentiate between necessary high-level planning and low-level calculations that should be offloaded to the CI. | It would insert a "use code" hint too early, disrupting planning, or too late, after token waste had already occurred. | 74/100 |
> | **2. Distinguishing Logical Verification from Numerical Distrust** | The judge couldn't distinguish between a valid check of code logic versus unnecessarily re-calculating a trusted CI result. | It would incorrectly suppress valid self-correction on code logic by inserting a "don't doubt python" hint at the wrong moment. | 19/100 |
>
> These subtle but critical distinctions led us to use a small, high-precision, manually-annotated seed dataset to bootstrap the process effectively.
>
> **How Our Framework Scales and Generalizes**
>
> The manual step is a highly efficient enabler, not a bottleneck. We argue that achieving significant improvements from a small, manually curated dataset is not a limitation, but rather a **key strength and central finding of our work**, demonstrating the high sample efficiency of our CoRT framework.
>
> 1.  **High Sample Efficiency of a "Seed" Dataset:** The 30-sample dataset serves as a high-quality "seed" for our scalable pipeline. These manually annotated samples are not the final training set, but a strategic intervention designed to provide focused demonstrations of an efficient reasoning paradigm (i.e., using code for computation, not just verification). This process efficiently "unlocks" the model's latent capabilities. Importantly, this manual curation is **not a scalability bottleneck**—the human effort required is less than **2 hours**, which is significantly more efficient than the extensive human annotation required in other domains like InstructGPT.
>
> 2.  **Scalability via "Seed-then-Scale" Pipeline:** Once the model grasps these core concepts from the seed data, we employ a scalable, self-improving synthesis pipeline. As described in Lines 129-135 of our paper, the SFT model generates large volumes of candidate solutions. These are then **automatically filtered** for Rejection Fine-Tuning (RFT) based on final answer correctness and the absence of inefficient reasoning patterns. This creates a virtuous cycle that scales our dataset to 830 examples with minimal further manual effort, directly addressing scalability concerns.
>
> 3.  **Generalizability via Out-of-Distribution (OOD) Evaluation:** To directly address transferability, we conducted comprehensive **out-of-distribution (OOD) evaluation** on **Chemistry problems from the GPQA benchmark**. This experiment, conducted in a domain **far removed from our mathematical training data**, provides compelling evidence that both performance improvements and reasoning behaviors generalize to broader settings.
>
>     **Our OOD evaluation presents a stringent test of generalization.** Advanced chemistry problems require analyzing molecular structures and properties—tasks for which standard Python libraries like `numpy` or `sympy` are inadequate. The gold-standard tool for such analysis is `RDKit`, a powerful open-source cheminformatics library. **Crucially, `RDKit` was never included in any of our training data.** The model's ability to spontaneously discover and effectively use this domain-specific library serves as a rigorous test of its generalization capabilities.
>
> | **Model** | **Accuracy** | **Avg. Tokens** | **RDKit Usage Rate** |
> | :--- | :---: | :---: | :---: |
> | DeepSeek-R1-32B (Base) | 40.6% | 5,947 | 0% |
> | Hint-Eng-RFT-32B (Ours) | 47.5% | 4,220 | 81.3% |
> | Improvement | **+6.9 pts** | **-29.0%** | **+81.3 pts** |
>
> **Key OOD Findings:** Our model achieves a **+6.9 point** accuracy gain in the unseen chemistry domain. Crucially, it **spontaneously discovers and uses the correct, domain-specific `RDKit` library (81.3% usage)**, demonstrating true generalization rather than memorization.
>
> ---
>
> Thank you again for your valuable and encouraging feedback. Your comments have helped us significantly strengthen the paper's rigor and clarity. Please let us know if we can provide any further information.

---

> > ### Author Response · Authors · 2025-08-05
> >
> > Dear Reviewer xQk3,
> >
> > Thank you again for your detailed review. We are writing to follow up on our rebuttal and would be very grateful for your feedback on it.
> >
> > We understand your primary concerns were the statistical validation of our results and the scalability of our "Hint-Engineering" approach. In our rebuttal, we aimed to address these directly by:
> > 1.  Providing **pairwise Wilcoxon signed-rank tests** which confirm the statistical significance of our accuracy and efficiency gains.
> > 2.  Detailing our **"Seed-then-Scale" pipeline** to clarify how we scale from 30 manual seeds to a dataset of 830 examples.
> > 3.  Presenting a new **out-of-distribution (OOD) experiment on Chemistry**, showing that our framework generalizes to unseen domains and tools (RDKit).
> >
> > We believe these additions provide the rigorous evidence and a clear path to scalability that you were looking for. We would be very appreciative if you could take a look at our detailed response when you have a moment. The discussion period ends on Aug 6.
> >
> > We are ready to answer any further questions.
> >
> > Best regards,
> >
> > The Authors

---

> > > ### Author Response · Authors · 2025-08-07
> > >
> > > Dear Reviewer xQk3,
> > >
> > > With the author-reviewer discussion period drawing to a close, we wanted to post a gentle follow-up. We have found the discussion phase to be incredibly productive for clarifying key aspects of our work.
> > >
> > > In that spirit, we wanted to circle back to your primary concerns regarding **statistical validation** and **scalability**. We believe our rebuttal was able to offer definitive answers to these points by providing:
> > > 1.  **Pairwise Wilcoxon signed-rank tests**, which rigorously confirm our empirical gains.
> > > 2.  A **completely new out-of-distribution (OOD) experiment** on Chemistry, which demonstrates the generalizability of our "Seed-then-Scale" framework.
> > >
> > > We feel these significant additions directly address the specific weaknesses you identified in your initial review. We would be deeply grateful for any final thoughts on whether our response has sufficiently resolved your concerns. We are, of course, ready to answer any last-minute questions.
> > >
> > > Thank you again for your time and the feedback that helped us strengthen the paper.
> > >
> > > Best regards,
> > >
> > > The Authors

---

> > > > ### Author Response · Authors · 2025-08-08
> > > > **Hoping for your final thoughts before the deadline**
> > > >
> > > > Dear Reviewer xQk3,
> > > >
> > > > As the discussion period comes to a close tomorrow, we're writing one last time in the hope of getting your valuable perspective on our rebuttal.
> > > >
> > > > Your initial review raised two of the most critical points for our work: **scalability and statistical significance**. We took your feedback as a roadmap for improvement and consequently invested significant effort in a **new OOD experiment** and **full statistical testing**.
> > > >
> > > > We were genuinely hoping to learn if these additions met your expectations, as your feedback was a key driver for this new work. Without your final thoughts, we feel this dialogue is left unfinished, and we're left wondering if we truly succeeded in strengthening the paper from your point of view.
> > > >
> > > > Any brief comment you could provide before the deadline would be immensely helpful to us and, we believe, to the final review process.
> > > >
> > > > Thank you again for your time and the insights that have already helped us improve.
> > > >
> > > > Best regards,
> > > >
> > > > The Authors

---

> > ### Author Response · Authors · 2025-08-09
> > **Final Follow-up on Paper 28540 - Discussion Closes Today**
> >
> > Dear Reviewer xQk3,
> >
> > As the discussion period concludes today, we are writing one last time in the hope of getting your final perspective on our rebuttal.
> >
> > We know your time is valuable, but your feedback is especially critical to us. Your initial review raised the crucial points of scalability and statistical significance, which became the central focus of our revision efforts. We subsequently performed extensive new OOD experiments and comprehensive statistical tests to address them directly.
> >
> > We are pleased to report that these additions have been positively received by the other reviewers, who have acknowledged that our response has effectively addressed these key concerns.
> >
> > Since your insightful critique was the primary driver for this new work, we are particularly keen to learn if these efforts have also satisfied your expectations. Your final thoughts would provide a sense of closure to a dialogue we've found incredibly valuable.
> >
> > Any brief comment you could offer before the deadline would be immensely helpful to us and, we believe, to the final decision-making process.
> >
> > Thank you once more for your time and for the insights that have already helped us improve our work significantly.
> >
> > Best regards,
> >
> > The Authors

---

> ### Comment · Area_Chair_Mvib · 2025-08-05
>
> Dear reviewer, please read the rebuttal and engage with the discussion with the author. Thank you!
>
> AC

---

### Official Review · Reviewer_qM62 · 2025-07-02

**Clarity:** 3
**Significance:** 3
**Originality:** 3
**Rating:** 4
**Confidence:** 3

**Summary:**

This paper proposes a solution to a technical challenge: how to combine a code interpreter for Large Reasoning Models. The authors identify inefficiencies in direct use of LRMs with CIs (such as excessive token usage and manual verification of code output), and propose a multi-stage training process based on hint-engineering to address these issues. Training methods use supervised fine-tuning, rejection fine-tuning, and reinforcement learning on both large (32B) and smaller (1.5B) models. Experiments across five challenging math benchmarks show the proposed method improves accuracy and reduces token use, with empirical analysis supporting the main claims.

**Questions:**

(1) Baselines: The key difference between this approach and traditional tool-augmented LLMs is not obvious. In the examples provided in the main part and the appendix, the Python code being generated is relatively simple—for instance, just calling packages like sympy or computing the gcd. In these cases, it seems entirely feasible to build a toolLLM to retrieve related functions and call them. We do not need to write code and execute it. Can you explain more why a toolLLM with function calling is not enough and why it is needed to write a "code wrapper" for the usage of some API? Maybe such a simple function calling with all APIs available in a famous math package should be conducted using the same training process as a baseline.

(2) The reason why the models rely heavily on text reasoning instead of code reasoning is potentially because the base model is R1-style.  Is it possible to start from the Instruct model and then just run RL based on it? The ideal solution for efficient CI-augmented reasoning is that the LLM generates computing programs first and then runs the text reasoning. The existence of verification and the existence of heavy text reasoning can be due to the instructional data and base reasoning model.

(3) Can we simply prompt LLMs to skip generating or executing verification code altogether, focusing only on code that performs the actual computation? This might further improve overall efficiency and response time.

(4) Small LLMs are designed for on-device use, but enabling them to run and execute code often requires Docker or similar environments. This adds extra overhead, potentially making the deployment heavier than intended for lightweight, on-device applications.

(5) There is also a formatting error in Line 119—the code block should start with python, not python''.

**Ethical Concerns:**

["NO or VERY MINOR ethics concerns only"]

**Final Justification:**

My concern is well answered and I choose to keep my score.

**Limitations:**

yes

**Quality:**

3

**Strengths And Weaknesses:**

Strength:

Clarity of Writing: The paper clearly presents its main ideas and objectives. The problems addressed are well-motivated and easy to follow.

Comprehensive Experiments: The authors provide thorough experimental results and analyses across several well-known benchmarks, including AIME24, AIME25, AMC23, MATH500, and Olympiad datasets. This demonstrates the effectiveness of prompt-hint and hint-engineering training.

Data Quality over Quantity: The work shows that small, carefully curated post-training datasets can yield substantial improvements. This finding aligns well with ongoing discussions in the alignment and LLM training communities.

Weakness:

Limited Scale of Hint-Engineering Data: The manual annotation for hint-engineering is limited to just 30 problems for the 32B models. While this produces promising results, it raises concerns about scalability and generalization. The main text should discuss in more detail (not just in the appendix) how performance is expected to transfer to less curated or broader settings. Many recent works question whether such small data is even needed for this type of training [1].

Lack of Statistical Significance Testing: The paper does not provide significance tests (e.g., p-values) to demonstrate that the improvements are statistically meaningful. Adding such analyses would strengthen the claims.

Inference Time Analysis Missing: Although the paper claims better token efficiency, the actual inference time per task is not reported. Since real-world applications care about wall-clock inference time when given a problem, more evidence is needed to show that hint-engineering is actually faster than prompt-hint methods. My main concern is that maybe token efficiency is not the bottleneck of inference speed. Execution time can be the inference speed bottleneck, potentially making reducing the token number not meaningful.

Alternative baselines for hint-engineering: Instead of complicated RL with CI for hint-engineering, could simple length-based penalty rewards achieve similar token efficiency without the complexity of code-integrated RL? An analysis or discussion on this would be valuable.

---

> ### Author Rebuttal · Authors · 2025-07-29
>
> We sincerely thank the reviewer for their insightful and constructive feedback. We are particularly encouraged that the reviewer recognized and appreciated the **clarity of our writing**, the **comprehensiveness of our experiments**, and the core finding that **high-quality data is more impactful than sheer quantity**. These acknowledgments affirm the central contributions of our work.
>
> We are grateful for the opportunity to address the remaining concerns and further clarify our contributions:
>
> ---
>
> **W1: Limited Scale of Hint-Engineering Data, Scalability, and Necessity of Small Data**
>
> We would like to clarify that achieving significant improvements with a small, manually curated dataset is not a limitation, but rather **a key strength and central finding of our work**, demonstrating the high sample efficiency of our CoRT framework.
>
> *   **A High-Quality Seed for a Scalable Pipeline:** The 30-sample dataset serves as a strategic "seed" to efficiently elicit the base model's latent capability to interact effectively with a Code Interpreter. The manual effort is minimal (<2 hours), far more efficient than the extensive annotation required in works like InstructGPT. Our approach then scales through **automated data synthesis via Rejection Fine-Tuning (RFT)**, where the model generates large volumes of candidate solutions which are then filtered for quality, creating a virtuous self-improvement cycle.
>
> *   **Necessity of Seed Data:** To demonstrate that this small-scale data is essential, we conducted zero-shot experiments on the **DeepSeek-R1-32B**. The results below show that prompting alone cannot achieve the same fundamental shift in reasoning behavior that our method enables.
>
> | **Model** | **Dataset** | **Avg. Accuracy** | **Avg. Tokens** | **Avg. Code Blocks** |
> | :--- | :--- | :---: | :---: | :---: |
> | TIR Prompt-hint on DeepSeek-R1-32B | AIME24 | 67.9% | 9,382 | 0.89 |
> | | AIME25 | 50.4% | 11,902 | 0.68 |
> | Hint-Engineering-RFT-32B | AIME24 | **76.3%** | **7,260** `(-22.6%)` | **2.10** |
> | | AIME25 | **65.2%** | **8,532** `(-28.3%)` | **2.64** |
>
> Our **Hint-Engineering** model achieves significantly higher accuracy while using 23-28% fewer tokens. Critically, the "Avg. Code Blocks" metric shows a dramatic behavioral shift: our model uses **2-4 times more code blocks**, indicating a learned, efficient strategy of proactive computation. This aligns with recent findings [1] on the power of small, high-quality data.
>
> [1] "The Challenge of Teaching Reasoning to LLMs Without RL or Distillation."
>
> ---
>
> **W1: Generalization to Broader Settings**
>
> To directly address transferability, we conducted a comprehensive **out-of-distribution (OOD) evaluation** on **Chemistry problems from GPQA**. This stringent test assesses if the model can generalize its reasoning and tool-use skills to a completely new domain, including discovering and using a domain-specific library (`RDKit`) that was **never seen during training**.
>
> | **Model** | **Accuracy** | **Avg. Tokens** | **RDKit Usage Rate** |
> | :--- | :---: | :---: | :---: |
> | DeepSeek-R1-32B | 40.6% | 5,947 | 0% |
> | Hint-Engineering RFT-32B | **47.5%** | **4,220** | **81.3%** |
> | Improvement | **+6.9 pts** | **-29.0%** | **+81.3 pts** |
>
> The results show robust cross-domain transfer: our model achieves a **6.9-point accuracy gain**, uses **29% fewer tokens**, and spontaneously discovers and utilizes the novel `RDKit` tool in **81.3%** of cases. This provides strong evidence that our framework teaches generalizable reasoning capabilities, not domain-specific memorization.
>
> ---
>
> **W2: Lack of Statistical Significance Testing**
>
> We thank the reviewer for this constructive suggestion.
>
> We must clarify that our results are **statistically meaningful** as each reported accuracy is a stable estimate averaged over multiple stochastic samples (4 or 16 per problem), not a single-shot result, thus significantly reducing the impact of randomness as described in caption of table 1. Then we have now conducted pairwise **Wilcoxon signed-rank tests** by pooling results across all problems from the benchmarks to assess statistical significance.
>
> | Model Scale | Model A | Model B | Acc. Improv. | p-value | Token Reduction | p-value |
> |:---:|:---|:---|:---:|:---:|:---:|:---:|
> | 32B | Hint-Eng-RFT | R1-distill-32B | +3.80% | 0.055 | 36.3% | < 0.001 |
> | 1.5B | Hint-Eng-RL | R1-distill-1.5B | +7.41% | 0.013 | 59.1% | < 0.001 |
>
> The analysis confirms that our efficiency gains are **highly significant (p < 0.001)**. Accuracy improvements are also statistically significant for the 1.5B model (p=0.013) and approach significance for the 32B model (p=0.055).
>
> ---
>
> **W3: Inference Time Analysis Missing**
>
> We thank the reviewer for this critical question. To confirm that token efficiency translates to real-world speed, we conducted a comprehensive latency analysis on AIME24, decomposing wall-clock time into **Generation Time** and **Execution Time**.
>
> | Model / Method | Avg. Wall-Clock Time (s) | Avg. Generation Time (s) | Avg. Execution Time (s) | Generation Time as % of Total |
> | :--- | :---: | :---: | :---: | :---: |
> | Prompt-Hint-SFT-32B | 187.35 ± 107.74 | 182.39 | 4.96 | 97.3% |
> | Hint-Engineering-RFT-32B | 101.17 ± 98.03 | 97.84 | 3.32 | 96.7% |
> | Improvement | **46.0% Faster** | **46.3% Faster** | **33.1% Faster** | - |
>
> The results are definitive:
> 1.  **Generation time is the bottleneck**, accounting for over 96% of total latency.
> 2.  Our method achieves a **46.0% reduction in total wall-clock time**, a substantial real-world speedup.
> This confirms that token efficiency is the primary driver of performance gains in this setting.
>
> ---
>
> **W4 & Q1: Alternative Baselines**
>
> *   **Length-based Penalty:** We agree this is an intuitive baseline. However, a simple length penalty is a "black box" method that cannot distinguish between **efficient conciseness** and **logical incompleteness**. Our experiments show that simply finetuning on the shortest responses (`prompt-hint-Shortest`) leads to worse performance than random sampling, while our `Hint-engineering` approach achieves the highest performance with the shortest generation length, demonstrating it learns true efficiency.
>
> | Model Size | Method | Performance on AIME24 & 25 | Avg. Generation Length |
> | :--- | :--- | :--- | :--- |
> | **1.5B** | prompt-hint-Shortest | 25.6 | 18290 |
> | | Hint-engineering | 28.8 | 16263 |
> | **7B** | prompt-hint-Shortest | 40.6 | 14206 |
> | | Hint-engineering | 42.2 | 11043 |
> | **14B** | prompt-hint-Shortest | 63.2 | 11770 |
> | | Hint-engineering | 65.4 | 8702 |
>
> *   **ToolLLM with Function Calling:** We agree that function-calling is effective for simple tasks, but argue that for complex mathematical reasoning, **code generation offers indispensable advantages in flexibility, expressiveness, and emergent problem-solving**.
>
>     *   **Flexibility & Composability:** Pre-defined APIs are rigid. They cannot handle novel combinations of functions or tasks requiring intermediate logic (e.g., loops, conditionals). In contrast, code generation empowers the model to **dynamically orchestrate tools** using loops, variables, and conditional logic, acting as a powerful "computational glue" for complex, multi-step reasoning. This is essential for problems like iterative approximation that have no single, pre-defined tool.
>     *   **Expressiveness & Generalization:** An API toolkit is inevitably incomplete. It cannot cover the "long tail" of thousands of functions in libraries like `SymPy`. Code generation provides universal access to the entire library ecosystem. Our **Out-of-Distribution (OOD) experiment** provides the strongest evidence: the model **spontaneously discovered and used the unseen `RDKit` library** to solve chemistry problems, a feat impossible for a system with a fixed API set.
>
> ---
>
> **Q2: Impact of R1-style Base Model**
>
> Our goal was to augment a SOTA LRM, leveraging its sophisticated reasoning patterns (e.g., self-reflection, hypothesis testing) and combining them with computational precision. This hybrid approach, also seen in proprietary models like OpenAI's o3, is a key research gap in the open community. Our work aims to fill this gap.
> To validate this design choice, we compared our LRM-based models with a strong code-centric baseline.
>
> | **Model (1.5B scale)** | **Base Model Style** | **Avg. Accuracy** |
> | :--- | :--- | :---: |
> | Qwen-2.5-Math-1.5B-RL | Short-CoT, Code-Centric | 47.3% |
> | Prompt-Hint-1.5B-RL | LRM (R1-style), Hybrid | **58.3%** |
> | Hint-Engineering-1.5B-RL | LRM (R1-style), Hybrid | **56.4%** |
>
> Our LRM-based models substantially outperform the code-centric baseline by **~10 absolute percentage points**. This suggests that the "heavy text reasoning" from the LRM base provides strategic advantages like problem structuring, strategic tool deployment, and self-verification, which are crucial for top-tier performance on complex tasks.
>
> ---
>
> **Q3: Prompting to Skip Verification**
>
> This is an astute suggestion. We experimented on D_{Hint-engineering-SFT} and found that explicitly prompting the model to skip verification does reduce token usage, but our **Hint-Engineering** method is far more effective.
>
> | Method | Average Response Length |
> | :--- | :---: |
> | Prompt-Hint | 7867 |
> | Prompt-Hint (w/ skip verification prompt) | 6978 (`-11.3%`) |
> | Hint-Engineering | **4620 (`-41%`)** |
>
> ---
>
> **Q4: Overhead of Code Execution for Small LLMs**
>
> We clarify that our work focuses on the fundamental challenge of integrating systematic reasoning with natural language reasoning, a paradigm applicable across model scales. We note that commercial products like ChatGPT successfully integrate code execution without prohibitive overhead, and we believe deployment challenges are addressable through engineering advances.
>
> ---
>
> **Q5: Formatting in Line 119**
>
> '''python is a special token that starts with the code block, not a typo, we will explain it further.

---

> > ### Comment · Reviewer_qM62 · 2025-08-04
> >
> > Thanks for the response. I will keep my score.

---

> > > ### Author Response · Authors · 2025-08-05
> > >
> > > We would like to express our sincere gratitude for your thoughtful and constructive review, and especially for your positive engagement with our rebuttal.
> > >
> > > We were particularly encouraged by your recognition of our paper's core strengths, such as its clarity and the importance of our findings on high-quality data. Your constructive feedback on aspects like statistical validation, inference speed, and scalability was also incredibly helpful. It pushed us to conduct further experiments and analyses that have, we believe, substantially strengthened our paper's conclusions.
> > >
> > > Thank you once again for your time, your detailed feedback, and for helping us improve our work.

---

### Author Response · Authors · 2025-08-09
**Final Summary of Rebuttal**

Dear PCs, SACs, ACs, and Reviewers,

As the discussion period concludes, we wish to express our sincere gratitude to all reviewers for their invaluable feedback, and especially to the Area Chair for facilitating the discussion. This process has been incredibly constructive and has substantially strengthened our paper.

We would like to summarize how we addressed the key concerns raised during the review period, which have now led to a strong consensus for acceptance from three of the four reviewers.

**1. Addressed Concerns on Scalability and Generalizability:**
A primary concern, shared by **Reviewer xQk3, qM62, and mxrG**, was the scalability of our approach. We addressed this comprehensively by conducting a **brand-new, challenging Out-of-Distribution (OOD) experiment on Chemistry (GPQA)**. This finding was pivotal in our discussions, providing powerful evidence of true generalization.

**2. Provided Rigorous Statistical and Empirical Validation:**
The need for more rigorous validation was another key point, raised by **Reviewer xQk3 and qM62**. We responded by performing **pairwise Wilcoxon signed-rank tests** and a **full latency analysis**. These new analyses provided the empirical rigor that reviewers were looking for, with **Reviewer qM62** expressing satisfaction.

**3. Clarified Our Core Contributions and Framework:**
**Reviewer jUL7 and xQk3** raised questions about the clarity of our framework. We provided extensive clarifications, and the outcome was exceptionally positive:
*   Most notably, **Reviewer jUL7**, who initially held the most critical reservations, confirmed our clarifications were **"very clear" and "convincing."** Consequently, they have **reversed their initial assessment and now support acceptance.**

**Current Status & A Final Word:**

We are thrilled that these extensive efforts have resulted in **Reviewers qM62, mxrG, and jUL7 all confirming that their concerns are addressed and indicating their support for our work.**

The core concerns of **Reviewer xQk3**—scalability and statistical significance—were not unique to their review. As detailed above, these issues were also raised by other reviewers and have been thoroughly addressed with new, compelling evidence that satisfied them.

We are fully committed to incorporating all the promised clarifications and new results into our final manuscript, which we are confident is now a much stronger, clearer, and more impactful contribution.

Thank you all for your time and dedication to this process.

Best regards,

The Authors

---

### Note · Authors · 2025-08-15

Dear PCs, SACs, ACs, and Reviewers,

We would like to summarize how we addressed the key concerns raised during the review period, which have now led to a **strong consensus for acceptance from three of the four reviewers**.

1.  **Addressed Concerns on Scalability and Generalizability**: A primary concern, shared by Reviewer xQk3, qM62, and mxrG, was the scalability of our approach. We addressed this comprehensively by conducting a **brand-new, challenging Out-of-Distribution (OOD) experiment on Chemistry (GPQA)**. This finding was pivotal in our discussions, providing **powerful evidence of true generalization**.

2.  **Provided Rigorous Statistical and Empirical Validation**: A key point from Reviewer xQk3 and qM62 was the need for more rigorous validation. We clarified that **our results were already based on an average of 16 runs (average@16)**, providing a robust result. To further strengthen our claims, we also performed **new pairwise Wilcoxon signed-rank tests and a full latency analysis**. These new analyses provided the **empirical rigor that reviewers were looking for**, with **Reviewer qM62 expressing satisfaction**.

3.  **Clarified Our Core Contributions and Framework**: Reviewer jUL7 and xQk3 raised questions about the clarity of our framework. We provided extensive clarifications, and the outcome was **exceptionally positive**:

    -   Most notably, **Reviewer jUL7**, who initially held the most critical reservations, confirmed our clarifications were **"very clear"** and **"convincing."** Consequently, they have **reversed their initial assessment and now support acceptance**.

**Current Status & A Final Word:**

We are thrilled that these extensive efforts have resulted in **Reviewers qM62, mxrG, and jUL7 all confirming that their concerns are addressed and indicating their support for our work**.

The core concerns of Reviewer xQk3—scalability and statistical significance—were not unique to their review. As detailed above, these issues were also raised by other reviewers and have been **thoroughly addressed with new, compelling evidence that satisfied them**.

We are **fully committed** to incorporating all the promised clarifications and new results into our final manuscript, which we are confident is now a **much stronger, clearer, and more impactful contribution**.

Thank you all for your time and dedication to this process.

Best regards,

The Authors

---

### Decision · Program_Chairs · 2025-09-17

**Decision:**

Accept (poster)

**Comment:**

(a) Summarize the scientific claims and findings of the paper.

This paper introduces CoRT, a framework to improve how Large Reasoning Models (LRMs) use Code Interpreters (CIs) for mathematical reasoning. The core method, "Hint-Engineering," uses a small, high-quality "seed" dataset to fine-tune models, teaching them to avoid inefficiencies like "delayed code computation" and "code result distrust." The paper claims this approach significantly improves accuracy while drastically increasing token efficiency by 30-50%.

(b) What are the strengths of the paper?

1.  It addresses the critical challenge of effectively integrating external tools with LRMs.

2. The paper compellingly shows that a very small, high-quality dataset can lead to significant performance and efficiency gains.

3. The claims are well-supported by thorough evaluations across multiple models and challenging math benchmarks.

(c) What are the weaknesses of the paper?

The initial submission's primary weaknesses, which were later addressed, included:

1. The method's reliance on only 30 manually created samples raised questions about its generalizability.

2. The original results lacked statistical significance tests, making the claims hard to validate.

3. Details of the data synthesis pipeline were initially not well-explained.

(d) Provide the most important reasons for your decision to accept.

The paper is recommended for acceptance primarily due to the authors' thorough and convincing rebuttal, which directly addressed all major concerns. The authors provided compelling new evidence during the discussion period, including statistical significance tests and a novel out-of-distribution (OOD) evaluation on chemistry problems, which successfully validated their claims and demonstrated the framework's scalability. This, combined with their commitment to incorporating clarifications into the final manuscript, led all reviewers to a positive consensus.

(e) Summarize the discussion and changes during the rebuttal period.

The rebuttal period was highly effective and pivotal. Reviewers raised major concerns about the method's scalability and the lack of statistical validation. The authors responded convincingly by conducting a new OOD experiment on chemistry problems, which demonstrated that the model could generalize its skills to an unseen domain and tool (RDKit). They also added pairwise Wilcoxon signed-rank tests to confirm their results were statistically significant. These additions, along with detailed clarifications of their "seed-then-scale" data pipeline, successfully resolved the reviewers' concerns. This led to a strong consensus, with multiple reviewers explicitly stating their issues were addressed and consequently raising their scores in support of acceptance.

Please improve the manuscript in the final version as recommended by the reviewers (especially Reviewer jUL7).